# Joint Enhancement and Classification using Coupled Diffusion Models of Signals and Logits

**Gilad Nurko**[1]  **Roi Benita**[1]  **Yehoshua Dissen**[1]  **Tomohiro Nakatani**[2]  **Marc Delcroix**[2]  **Shoko Araki**[2]
**Joseph Keshet**[1]

## Abstract

Robust classification in noisy environments remains a fundamental challenge in machine learning. Standard approaches typically treat signal enhancement and classification as separate, sequential stages: first enhancing the signal and then applying a classifier. This approach fails to leverage the semantic information in the classifier's output during denoising. In this work, we propose a general, domain-agnostic framework that integrates two interacting diffusion models: one operating on the input signal and the other on the classifier's output logits, without requiring any retraining or fine-tuning of the classifier. This coupled formulation enables mutual guidance, where the enhancing signal refines the class estimation and, conversely, the evolving class logits guide the signal reconstruction towards discriminative regions of the manifold. We introduce three strategies to effectively model the joint distribution of the input and the logit. We evaluated our joint enhancement method for image classification and automatic speech recognition. The proposed framework surpasses traditional sequential enhancement baselines, delivering robust and flexible improvements in classification accuracy under diverse noise conditions.

## 1. Introduction

Recent studies have demonstrated that while standard deep classifiers excel in ideal conditions, their performance degrades catastrophically under even mild distribu-

[0]Code is available at https://github.com/gilad-nurko/coupled-diffusion.git.

[1]Technion – Israel Institute of Technology, Haifa, Israel [2]NTT, Inc., Japan. Correspondence to: Gilad Nurko <gilad.nurko@campus.technion.ac.il>.

*Proceedings of the 43rd International Conference on Machine Learning*, Seoul, South Korea. PMLR 306, 2026. Copyright 2026 by the author(s).

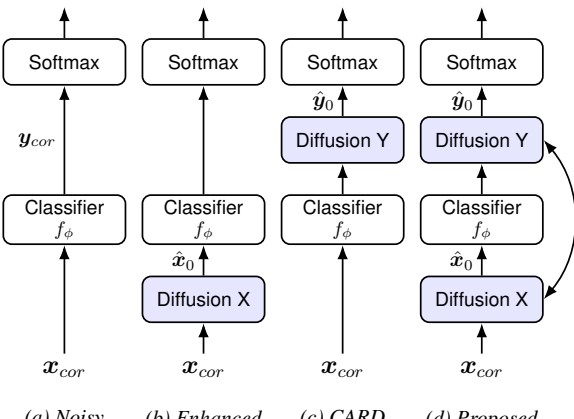

*Figure 1.* Comparison of noisy signal classification paradigms. $x_{cor}$, $y_{cor}$ denote the corrupted signal and logits; $\hat{x}_0$, $\hat{y}_0$ their denoised versions. (a) Noisy: direct classification. (b) Enhanced: independent signal enhancement followed by classification. (c) CARD: denoising logits, $y$, given fixed corrupted signal, $x_{cor}$. (d) Proposed: coupled denoising of both signal and logits.

tion shifts or input corruptions (Hendrycks & Dietterich, 2019; Geirhos et al., 2018; Recht et al., 2019).

A common paradigm to address this challenge is the sequential "enhance-then-classify" approach, where a dedicated enhancement module is employed to remove noise before feeding the signal into a standard classifier (Weninger et al., 2015; Liu et al., 2022a). This paradigm is depicted conceptually in Fig. 1(b) using a diffusion model-based enhancement module. While intuitively appealing, this decoupling introduces a significant misalignment between the objectives of the two stages. Enhancement models are typically optimized for perceptual quality or signal-level reconstruction metrics, such as Mean Squared Error (MSE), Signal-to-Noise Ratio (SNR) or Perceptual Evaluation of Speech Quality (PESQ) (Haeb-Umbach et al., 2020), which do not necessarily correlate with classification accuracy (Blau & Michaeli, 2018; Diamond et al., 2021; Loizou, 2007). In fact, recent analyses have shown that aggressive denoising can introduce processing artifacts that are perceptually subtle but highly disruptive to the downstream classifier, often causing more harm than the

original noise itself (Sun et al., 2022; Wang et al., 2020; Iwamoto et al., 2022).

Another potential remedy is to adapt the classifier directly to the noisy domain via fine-tuning or multi-style training on corrupted data. State-of-the-art classifiers and foundation models are trained on massive amount of data. Adapting these models to a specific noise profile frequently induces catastrophic forgetting (Kirkpatrick et al., 2017; Kemker et al., 2018), harming performance on clean data and unseen corruptions. Recent theory further suggests that fine-tuning on noisy or limited data distorts pre-trained robust representations, resulting in *overfitting* and poor generalization (Kumar et al., 2022; Wortsman et al., 2022).

Modeling both the input signal and the label was first proposed using Energy-Based Models (EBMs). Joint Energy-Based Model (JEM) framework (Grathwohl et al., 2019) proposed to reinterpret the logits of a standard classifier to define an unnormalized density $p(\boldsymbol{x}, \boldsymbol{y})$. By training the model to maximize the joint likelihood $p(\boldsymbol{x}, \boldsymbol{y})$ rather than just the conditional $p(\boldsymbol{y}|\boldsymbol{x})$, JEM achieves improved robustness and out-of-distribution detection compared to standard classifiers. However, this method suffers from training instabilities and imposes significant training and architectural constraints, including the need to retrain the classifier itself.

In the specific context of automatic speech recognition (ASR), robustness has been typically tackled via front-end enhancement (Donahue et al., 2018) or multi-style training (Kinoshita et al., 2020). As noted in the introduction, enhancement modules often introduce artifacts detrimental to ASR (Iwamoto et al., 2022). Several recent works therefore propose end-to-end optimization of enhancement using ASR objectives, by jointly training the enhancement and recognition models (Subramanian et al., 2019; Chang et al., 2022). Such approaches, however, require training the ASR model itself, which becomes increasingly impractical for modern large-scale systems. To address this challenge, Dissen et al. (2025) keep the ASR model frozen and instead introduce a learnable adapter trained with the frozen ASR model's loss together with a small enhancement loss as regularization, with a conceptually related idea explored in vision by Son et al. (2020).

In this work, we propose a unified framework that bridges the gap between signal enhancement and generative classification by treating both the signal reconstruction and the label prediction as coupled diffusion processes, and avoid any change (including fine-tuning) to the classifier weights. Our work was inspired by Classification and Regression Diffusion (CARD) (Han et al., 2022) that proposes a diffusion model to denoise the logits of a pre-trained classifier as depicted in Fig. 1(c). CARD's main limitation is its inability to generalize to noisy inputs, as the logits denoiser relies solely on the corrupted signal and therefore loses valuable information present in the underlying clean input.

We jointly train two diffusion models to denoise the input signal and the classifier's logits simultaneously (see Fig. 1(d)). Central to our approach is *mutual guidance*, a process where the two models inform one another: the signal provides the logit model with accurate features, and the logits guide the signal reconstruction toward more semantically meaningful areas of the data manifold. This iterative interaction is analogous to alternating minimization procedures (Csiszár & Tusnády, 1984), in which latent signal and latent logit estimates are dynamically refined through mutual updates. Within this framework, we investigate three interaction strategies between the signal and logit diffusion models, namely, *Parallel*, *Alternating*, and *Nested*, which offer flexible trade-offs between computational efficiency and the granularity of mutual guidance.

We formalize our approach as a general framework applicable to classification tasks involving high-dimensional data. To demonstrate its practical effectiveness, we evaluate our method on both image classification and ASR tasks. Our experimental results show that enabling interaction between the signal and label diffusion processes substantially improves robustness over both enhancement-based baselines and static conditional generation methods (Han et al., 2022) across multiple datasets. While the experiments on image classifications serve as a proof of concept, ASR experiments are carried out using the powerful Whisper model (Radford et al., 2023), widely used both for research and commercial applications, and thus show the potential of our proposed method to improve strong systems on several noisy speech benchmarks.

## 2. Preliminaries

Before presenting our approach, we briefly review the diffusion model framework. Although the proposed method is not restricted to a particular formulation, we describe it in the context of denoising diffusion probabilistic models (DDPM) (Ho et al., 2020; Sohl-Dickstein et al., 2015). Empirically, in Sec. 4, we evaluate the method using DDPM, denoising diffusion implicit models (DDIM) (Song et al., 2020a), and stochastic differential equation (SDE)–based formulations (Song et al., 2020b).

DDPMs are generative models that learn to generate samples $\boldsymbol{x}_0$ from an unknown data distribution $q(\boldsymbol{x}_0)$ by reversing a gradual noising process. The forward process is a fixed Markov chain of length $T$ that gradually adds Gaussian noise to the data according to a variance schedule $\beta_1, \ldots, \beta_T$:

$$q(\boldsymbol{x}_t \mid \boldsymbol{x}_{t-1}) = \mathcal{N}(\boldsymbol{x}_t; \sqrt{1 - \beta_t}\boldsymbol{x}_{t-1}, \beta_t\boldsymbol{I}), \quad (1)$$

where $t \in [1, T]$, $t = 0$ corresponds to clean data, and

$t = T$ to pure Gaussian noise.

The generative process is defined by the reverse Markov chain, $p_\theta(\boldsymbol{x}_{t-1} \mid \boldsymbol{x}_t)$, which is modeled by a neural network with parameters $\theta$. A key property of diffusion models is that the true posterior induced by the forward process, conditioned on the clean data $\boldsymbol{x}_0$, is tractable and Gaussian:

$$q(\boldsymbol{x}_{t-1} \mid \boldsymbol{x}_t, \boldsymbol{x}_0) = \mathcal{N}(\boldsymbol{x}_{t-1}; \tilde{\boldsymbol{\mu}}(\boldsymbol{x}_t, \boldsymbol{x}_0), \tilde{\beta}_t \boldsymbol{I}), \quad (2)$$

where $\tilde{\boldsymbol{\mu}}$ is a linear combination of $\boldsymbol{x}_t$ and $\boldsymbol{x}_0$ , and $\tilde{\beta}_t$ is the corresponding posterior variance at timestep $t$, both determined by the diffusion method. This implies that the reverse transition is analytically available given the clean sample.

In practice, $\boldsymbol{x}_0$ is unknown and must be replaced by a learned estimate. Following Ho et al. (2020), all reverse distributions $p_\theta(\cdot)$, including $p_\theta(\boldsymbol{x}_0|\boldsymbol{x}_t)$ and $p_\theta(\boldsymbol{x}_{t-1}|\boldsymbol{x}_t)$, are implemented via a denoising network "Denoiser$_\theta$" that predicts the forward diffusion noise $\boldsymbol{\epsilon}$ used to transform $\boldsymbol{x}_0$ into $\boldsymbol{x}_t$. The predicted noise, the estimated clean sample $\hat{\boldsymbol{x}}_0^{(t)}$, and $\boldsymbol{x}_{t-1}$ are in one-to-one correspondence: each of them uniquely determines the others through the diffusion equations. The reverse transition is obtained by substituting $\hat{\boldsymbol{x}}_0^{(t)}$ into the analytic posterior

$$p_\theta(\boldsymbol{x}_{t-1} \mid \boldsymbol{x}_t) := q(\boldsymbol{x}_{t-1} \mid \boldsymbol{x}_t, \hat{\boldsymbol{x}}_0^{(t)}). \quad (3)$$

Following Elata et al. (2024), we compute $\hat{\boldsymbol{x}}_0^{(t)}$ and then derive $\boldsymbol{x}_{t-1}$ from it. This is equivalent to the DDPM formulation and is adopted for convenience, as $\hat{\boldsymbol{x}}_0^{(t)}$ serves as a useful intermediate in our coupled framework, where the clean-data estimate produced by one diffusion model is used to condition the denoiser of the other.

Note that we introduced here the diffusion process for the input signal $\boldsymbol{x}$, but the same formulation can also be used for a diffusion process over the logits, $\boldsymbol{y}$.

## 3. Method

We start by presenting the problem setting and notation. We then present a framework that defines the interaction between the signal and the logits diffusion processes. We conclude by deriving three strategies that instantiate this framework: *Parallel*, *Alternating*, and *Nested*.

### 3.1. Problem Formulation and Notation

We consider a classification setting in which we observe a corrupted input signal $\boldsymbol{x}_{cor}$, derived from an unknown clean signal $\boldsymbol{x}_0$. Instead of modeling discrete class labels directly, we define our target $\boldsymbol{y}_0$ as the continuous clean class logits, i.e., the pre-softmax class score vectors output by a classifier. This continuous formulation en-

ables gradient-based diffusion modeling while preserving semantic information.

We assume access to a pre-trained classifier $f_\phi$ that maps a signal to logits. The classifier parameters $\phi$ remain frozen by design and are not updated during the process. In particular, we define the clean logits as $\boldsymbol{y}_0 = f_\phi(\boldsymbol{x}_0)$, and the logits of the corrupted observation as $\boldsymbol{y}_{cor} = f_\phi(\boldsymbol{x}_{cor})$. Throughout this work, we use conditioning of the logit diffusion process $\boldsymbol{y}$ on a signal $\boldsymbol{x}$ (i.e., $\boldsymbol{x}_{cor}$, $\boldsymbol{x}_t$, or $\boldsymbol{x}_{t-1}$) to denote conditioning on both the signal and its classifier output $f_\phi(\boldsymbol{x})$. Together, these sources provide complementary raw and semantic information that guides the process.

Our objective is to robustly estimate the clean logits $\boldsymbol{y}_0$ from the corrupted observation $\boldsymbol{x}_{cor}$ by modeling a reverse diffusion process over latent variables conditioned on the corrupted input. To this end, we employ two coupled diffusion processes: one operating on the corrupted signal and the other on the noisy logits. Both processes are indexed by timestep $t \in [0, T]$. Specifically, we define: (i) the signal through the diffusion process $\{\boldsymbol{x}_t\}_{t=0}^{T}$ that generates the clean signal, and (ii) the logits through the diffusion process $\{\boldsymbol{y}_t\}_{t=0}^{T}$ that generates the clean class logits. During inference, both processes begin from their respective noise distributions, $\boldsymbol{x}_T \sim \mathcal{N}(0, \boldsymbol{I})$ and $\boldsymbol{y}_T \sim \mathcal{N}(0, \boldsymbol{I})$, and are jointly denoised through iterative refinement.

We adopt the following notational conventions: $p_\theta(\cdot)$ denotes learned reverse diffusion distributions parameterized by neural networks with parameters $\theta$. We use this notation both for one-step reverse transitions, such as $p_\theta(\boldsymbol{x}_{t-1} \mid \cdot)$, and for clean-sample prediction distributions, such as $p_\theta(\boldsymbol{x}_0 \mid \cdot)$. In the latter case, $p_\theta(\boldsymbol{x}_0 \mid \cdot)$ is implemented deterministically: it is interpreted as a Dirac delta distribution centered at the neural network prediction $\hat{\boldsymbol{x}}_0$. The notation $q(\cdot)$ denotes the fixed, tractable Gaussian posteriors induced by the forward diffusion schedule. We use $p_\theta^{(x)}(\cdot)$ and $p_\theta^{(y)}(\cdot)$ to denote the signal and logits instantiations of $p_\theta$, respectively.

### 3.2. Generalized Coupled Diffusion Framework

Unlike standard sequential approaches that first enhance the signal and then classify independently, our method formulates a coupled generative process where signal and logit estimation mutually inform each other. The estimated clean signal $\hat{\boldsymbol{x}}_0^{(t)}$ provides semantic context for logit generation, while the evolving class predictions $\hat{\boldsymbol{y}}_0^{(t)}$ impose structural constraints that guide signal reconstruction.

To realize this bidirectional coupling, we propose to execute two diffusion processes that condition on each other, while a scheduler determines the order in which they are

**Algorithm 1** Generalized Coupled Diffusion

1: **Input:** $\boldsymbol{x}_{cor}$, scheduler $\mathcal{T}$
2: Initialize $\boldsymbol{x}_T \sim \mathcal{N}(0, \boldsymbol{I})$, $\boldsymbol{y}_T \sim \mathcal{N}(0, \boldsymbol{I})$
3: $\hat{\boldsymbol{y}}_{\text{cond}} \leftarrow f_\phi(\boldsymbol{x}_{cor}), \quad \hat{\boldsymbol{x}}_{\text{cond}} \leftarrow \boldsymbol{x}_{cor}$
4: **for** each action in scheduler $\mathcal{T}$ **do**
5:      **if** action = "update $\boldsymbol{x}$" **then**
6:          $\hat{\boldsymbol{x}}_0^{(t)} \sim p_\theta^{(x)}(\boldsymbol{x}_0 \mid \boldsymbol{x}_t, \hat{\boldsymbol{y}}_{\text{cond}}, \boldsymbol{x}_{cor})$
7:          $\boldsymbol{x}_{t-1} \sim q\left(\boldsymbol{x}_{t-1} \mid \boldsymbol{x}_t, \hat{\boldsymbol{x}}_0^{(t)}\right)$
8:          Update $\hat{\boldsymbol{x}}_{\text{cond}}$ if specified by $\mathcal{T}$
9:      **else if** action = "update $\boldsymbol{y}$" **then**
10:         $\hat{\boldsymbol{y}}_0^{(t)} \sim p_\theta^{(y)}(\boldsymbol{y}_0 \mid \boldsymbol{y}_t, \hat{\boldsymbol{x}}_{\text{cond}}, \boldsymbol{x}_{cor})$
11:         $\boldsymbol{y}_{t-1} \sim q\left(\boldsymbol{y}_{t-1} \mid \boldsymbol{y}_t, \hat{\boldsymbol{y}}_0^{(t)}\right)$
12:         Update $\hat{\boldsymbol{y}}_{\text{cond}}$ if specified by $\mathcal{T}$
13:      **end if**
14: **end for**
15: **return** $\boldsymbol{y}_0$

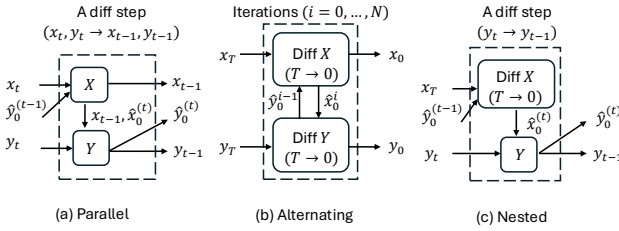

(a) Parallel     (b) Alternating     (c) Nested

*Figure 2.* Overview of the proposed architectures. (a) *Parallel*: $\boldsymbol{x}$ and $\boldsymbol{y}$ are denoised simultaneously, exchanging estimates $\hat{\boldsymbol{x}}_0$ and $\hat{\boldsymbol{y}}_0$ at each timestep $t$ for step-wise cross-conditioning. (b) *Alternating*: an iterative refinement strategy where complete diffusion trajectories for $\boldsymbol{x}$ and $\boldsymbol{y}$ condition each other in alternating turns. (c) *Nested*: a guidance mechanism where every single step of the outer $\boldsymbol{y}$ process triggers a full inner diffusion loop over $\boldsymbol{x}$.

applied. We maintain two coupled transition distributions:

$$\boldsymbol{x}_{t-1} \sim p_\theta^{(x)}(\boldsymbol{x}_{t-1} \mid \boldsymbol{x}_t, \hat{\boldsymbol{y}}_{\text{cond}}, \boldsymbol{x}_{cor}), \quad (4)$$

$$\boldsymbol{y}_{t-1} \sim p_\theta^{(y)}(\boldsymbol{y}_{t-1} \mid \boldsymbol{y}_t, \hat{\boldsymbol{x}}_{\text{cond}}, \boldsymbol{x}_{cor}), \quad (5)$$

where $\hat{\boldsymbol{x}}_{\text{cond}}$ and $\hat{\boldsymbol{y}}_{\text{cond}}$ are conditioning estimates derived from the other process. A scheduler $\mathcal{T}$ determines the execution order: at each iteration, it decides whether to perform a diffusion step on $\boldsymbol{x}$ or on $\boldsymbol{y}$, and when to update the conditioning variables. This generalized procedure is summarized in Algo. 1.

The scheduler $\mathcal{T}$ can be designed to express the interactions between the joint diffusion models, and can be instantiated in many ways. In this work, we focus on three implementations: *Parallel*, *Alternating*, and *Nested*, each corresponding to a distinct scheduler with different trade-offs between coupling strength and computational cost.

## 3.3. Parallel Strategy

The *Parallel* strategy (Fig. 2a) uses an *interleaved scheduler*. It performs a single joint diffusion loop in which each timestep alternates between a signal update and a logit update, with each update guided by the most recent prediction of the other process. This design is the most direct and computationally efficient realization of mutual guidance.

**Joint reverse-time factorization.** At inference, we aim to sample from the joint reverse distribution, by exploiting the Markov structure of the reverse diffusion process,

$$p(\boldsymbol{x}_{0:T-1}, \boldsymbol{y}_{0:T-1} \mid \boldsymbol{x}_T, \boldsymbol{y}_T, \boldsymbol{x}_{cor}) = \\ \prod_{t=1}^T p(\boldsymbol{x}_{t-1}, \boldsymbol{y}_{t-1} \mid \boldsymbol{x}_t, \boldsymbol{y}_t, \boldsymbol{x}_{cor}), \quad (6)$$

where $\boldsymbol{x}_T \sim \mathcal{N}(0, I)$ and $\boldsymbol{y}_T \sim \mathcal{N}(0, I)$. Applying the chain rule to each joint transition yields

$$p(\boldsymbol{x}_{0:T-1}, \boldsymbol{y}_{0:T-1} \mid \boldsymbol{x}_T, \boldsymbol{y}_T, \boldsymbol{x}_{cor}) = \\ \prod_{t=1}^T p(\boldsymbol{x}_{t-1} \mid \boldsymbol{x}_t, \boldsymbol{y}_t, \boldsymbol{x}_{cor}) p(\boldsymbol{y}_{t-1} \mid \boldsymbol{x}_{t-1}, \boldsymbol{x}_t, \boldsymbol{y}_t, \boldsymbol{x}_{cor}). \quad (7)$$

This induces a natural update order in which $\boldsymbol{x}_{t-1}$ is first sampled conditioned on the current signal and logit states, followed by sampling $\boldsymbol{y}_{t-1}$ conditioned on the updated signal. We now derive tractable approximations for each of these conditional distributions.

**Signal update.** We first consider the signal transition. Using marginalization we get:

$$p(\boldsymbol{x}_{t-1} \mid \boldsymbol{x}_t, \boldsymbol{y}_t, \boldsymbol{x}_{cor}) = \\ \int p(\boldsymbol{x}_{t-1} \mid \boldsymbol{x}_t, \boldsymbol{x}_0, \boldsymbol{y}_t, \boldsymbol{x}_{cor}) p(\boldsymbol{x}_0 \mid \boldsymbol{x}_t, \boldsymbol{y}_t, \boldsymbol{x}_{cor}) \, d\boldsymbol{x}_0 \quad (8)$$

This integral is generally intractable. Following standard practice in diffusion models (Ho et al., 2020), we approximate the posterior over $\boldsymbol{x}_0$ using a point estimate produced by a neural denoiser. In our coupled setting, we additionally condition this estimate on the most recent prediction of the clean logits, which is available from the previous timestep and provides complementary semantic information. Concretely, we adopt the approximation

$$p(\boldsymbol{x}_0 \mid \boldsymbol{x}_t, \boldsymbol{y}_t, \boldsymbol{x}_{cor}) \approx \delta\left(\boldsymbol{x}_0 - \hat{\boldsymbol{x}}_0^{(t)}\right), \quad (9)$$

where the clean signal estimate is obtained as

$$\hat{\boldsymbol{x}}_0^{(t)} \sim p_\theta^{(x)}\left(\boldsymbol{x}_0 \mid \boldsymbol{x}_t, \hat{\boldsymbol{y}}_0^{(t+1)}, \boldsymbol{y}_t, \boldsymbol{x}_{cor}\right). \quad (10)$$

Substituting this approximation into (8) collapses the integral, yielding

$$p(\boldsymbol{x}_{t-1} \mid \boldsymbol{x}_t, \boldsymbol{y}_t, \boldsymbol{x}_{cor}) \approx q(\boldsymbol{x}_{t-1} \mid \boldsymbol{x}_t, \hat{\boldsymbol{x}}_0^{(t)}, \boldsymbol{y}_t, \boldsymbol{x}_{cor}),$$

Importantly, conditioned on $(\boldsymbol{x}_t, \hat{\boldsymbol{x}}_0^{(t)})$, the Gaussian posterior $q(\boldsymbol{x}_{t-1}|\cdot)$ is independent of $(\boldsymbol{y}_t, \boldsymbol{x}_{cor})$. These variables influence the transition only through $\hat{\boldsymbol{x}}_0^{(t)}$ and are therefore redundant in the conditioning. Formally,

$$q(\boldsymbol{x}_{t-1} \mid \boldsymbol{x}_t, \hat{\boldsymbol{x}}_0^{(t)}, \boldsymbol{y}_t, \boldsymbol{x}_{cor}) = q(\boldsymbol{x}_{t-1} \mid \boldsymbol{x}_t, \hat{\boldsymbol{x}}_0^{(t)}), \quad (11)$$

see Appendix A in the Supplementary for details. This results in a two-step update at each timestep: first, predict $\hat{\boldsymbol{x}}_0^{(t)}$, then sample $\boldsymbol{x}_{t-1}$ from the corresponding posterior.

**Logit update.** We now turn to the logit transition. Since $\boldsymbol{x}_{t-1}$ is a denoised refinement of $\boldsymbol{x}_t$, we treat conditioning on $\boldsymbol{x}_t$ as redundant and assume

$$p(\boldsymbol{y}_{t-1} \mid \boldsymbol{x}_{t-1}, \boldsymbol{x}_t, \boldsymbol{y}_t, \boldsymbol{x}_{cor}) = p(\boldsymbol{y}_{t-1} \mid \boldsymbol{x}_{t-1}, \boldsymbol{y}_t, \boldsymbol{x}_{cor}).$$

Analogously, we marginalize over the clean logits $\boldsymbol{y}_0$:

$$p(\boldsymbol{y}_{t-1} \mid \boldsymbol{x}_{t-1}, \boldsymbol{y}_t, \boldsymbol{x}_{cor}) = \\ \int p(\boldsymbol{y}_{t-1} \mid \boldsymbol{y}_t, \boldsymbol{y}_0, \boldsymbol{x}_{t-1}, \boldsymbol{x}_{cor}) \\ p(\boldsymbol{y}_0 \mid \boldsymbol{y}_t, \boldsymbol{x}_{t-1}, \boldsymbol{x}_{cor}) \, d\boldsymbol{y}_0. \quad (12)$$

We again apply a point-estimate approximation,

$$p(\boldsymbol{y}_0 \mid \boldsymbol{y}_t, \boldsymbol{x}_{t-1}, \boldsymbol{x}_{cor}) \approx \delta\left(\boldsymbol{y}_0 - \hat{\boldsymbol{y}}_0^{(t)}\right), \quad (13)$$

where the clean logit estimate is predicted as

$$\hat{\boldsymbol{y}}_0^{(t)} \sim p_\theta^{(y)}\left(\boldsymbol{y}_0 \mid \boldsymbol{y}_t, \hat{\boldsymbol{x}}_0^{(t)}, \boldsymbol{x}_{t-1}, \boldsymbol{x}_{cor}\right). \quad (14)$$

Substituting into (12) yields the tractable distribution

$$p(\boldsymbol{y}_{t-1} \mid \boldsymbol{x}_{t-1}, \boldsymbol{y}_t, \boldsymbol{x}_{cor}) \approx q\left(\boldsymbol{y}_{t-1} \mid \boldsymbol{y}_t, \hat{\boldsymbol{y}}_0^{(t)}, \boldsymbol{x}_{t-1}, \boldsymbol{x}_{cor}\right) \\ = q\left(\boldsymbol{y}_{t-1} \mid \boldsymbol{y}_t, \hat{\boldsymbol{y}}_0^{(t)}\right), \quad (15)$$

where the last equality follows as in the signal update (see Appendix A).

**Resulting algorithm.** The derivation above induces a joint reverse-time sampling procedure in which signal and logits are updated sequentially within each timestep, each conditioned on the most recent estimate of the other. The complete procedure is summarized in Algo. 2.

**Training objective.** Training minimizes a joint noise-prediction objective summing signal and logit losses. Crucially, to enable effective mutual guidance, we explicitly simulate cross-conditioning during optimization: intermediate estimates of the clean signal and logits are generated on-the-fly to condition the respective denoisers. This

---

**Algorithm 2** Parallel Inference Algorithm

1: $\boldsymbol{y}_T \sim \mathcal{N}(0, \boldsymbol{I})$
2: $\boldsymbol{x}_T \sim \mathcal{N}(0, \boldsymbol{I})$
3: $\hat{\boldsymbol{y}}_0^{(T+1)} \leftarrow f_\phi(\boldsymbol{x}_{cor})$
4: **for** $t = T$ **to** $1$ **do**
5:    $\hat{\boldsymbol{x}}_0^{(t)} \sim p_\theta^{(x)}(\boldsymbol{x}_0 \mid \boldsymbol{x}_t, \boldsymbol{y}_t, \hat{\boldsymbol{y}}_0^{(t+1)}, \boldsymbol{x}_{cor})$
6:    $\boldsymbol{x}_{t-1} \sim q(\boldsymbol{x}_{t-1} \mid \boldsymbol{x}_t, \hat{\boldsymbol{x}}_0^{(t)})$
7:    $\hat{\boldsymbol{y}}_0^{(t)} \sim p_\theta^{(y)}(\boldsymbol{y}_0 \mid \boldsymbol{y}_t, \boldsymbol{x}_{t-1}, \hat{\boldsymbol{x}}_0^{(t)}, \boldsymbol{x}_{cor})$
8:    $\boldsymbol{y}_{t-1} \sim q(\boldsymbol{y}_{t-1} \mid \boldsymbol{y}_t, \hat{\boldsymbol{y}}_0^{(t)})$
9: **end for**
10: **return** $\boldsymbol{y}_0$

---

aligns the training distribution with the inference dynamics described above. Notably, training does not require ground-truth labels, but relies solely on the clean signal corresponding to the corrupted input. Recall that $p_\theta(\cdot)$ does not introduce a separate model: all reverse transitions are implemented via the same denoising network Denoiser$_\theta$ (Sec. 2). The procedure is detailed in Algo. 3.

### 3.4. Alternating Strategy

The *Alternating* strategy (Fig. 2(b)) uses a *block scheduler*. It decouples the two processes temporally by executing a full diffusion trajectory for the signal, followed by a full diffusion trajectory for the logits, and repeating this procedure for multiple iterations. Each process is guided by a fully denoised sample of the other from the previous iteration. This aligns with standard diffusion pipelines by relying on clean estimates rather than intermediate states.

We denote by $\boldsymbol{x}_t^i$ and $\boldsymbol{y}_t^i$ the noisy signal and logits at diffusion timestep $t$ during iteration $i$. Their fully denoised estimates at $t = 0$ are denoted $\hat{\boldsymbol{x}}_0^i$ and $\hat{\boldsymbol{y}}_0^i$, respectively. These point estimates are treated as deterministic conditioning variables in subsequent iterations.

**Proposition 3.1.** *Given a corrupted observation $\boldsymbol{x}_{cor}$ and a logit estimate $\hat{\boldsymbol{y}}_0^{i-1}$ from the previous iteration, the* Alternating *strategy approximately maximizes the joint conditional probability, at each iteration $i$.*

$$p(\boldsymbol{x}_{0:T-1}, \boldsymbol{y}_{0:T-1} | \boldsymbol{x}_T, \boldsymbol{y}_T, \boldsymbol{x}_{cor}) \approx \\ \prod_{t=1}^{T} p(\boldsymbol{x}_{t-1} | \boldsymbol{x}_t, \hat{\boldsymbol{y}}_0^{i-1}, \boldsymbol{x}_{cor}) \prod_{t=1}^{T} p(\boldsymbol{y}_{t-1} | \boldsymbol{y}_t, \hat{\boldsymbol{x}}_0^i, \boldsymbol{x}_{cor}),$$

*where $\hat{\boldsymbol{x}}_0^i$ is the clean signal point estimate obtained at the terminal step ($t = 0$) of the signal diffusion trajectory.*

Further motivation, the inference algorithm, a detailed probabilistic derivation, and training considerations are provided in Appendix B.

**Algorithm 3** Parallel Training Algorithm

1: **repeat** until convergence
2: Sample $(\boldsymbol{x}_0, \boldsymbol{x}_{cor})$ from $\mathcal{D}$
3: $\boldsymbol{y}_0 \leftarrow f_\phi(\boldsymbol{x}_0)$
4: Sample $t \sim \text{Uniform}(\{1, \dots, T\})$
5: Sample $\boldsymbol{\epsilon}_x, \boldsymbol{\epsilon}_y \sim \mathcal{N}(0, \boldsymbol{I})$
6: $\hat{\boldsymbol{y}}_0^{(t)} \leftarrow f_\phi(\boldsymbol{x}_{cor})$
7: $\boldsymbol{x}_t \sim q(\boldsymbol{x}_t \mid \boldsymbol{x}_0, \boldsymbol{\epsilon}_x), \boldsymbol{y}_t \sim q(\boldsymbol{y}_t \mid \boldsymbol{y}_0, \boldsymbol{\epsilon}_y)$
8: $\hat{\boldsymbol{x}}_0^{(t)} \sim p_\theta^{(x)}(\boldsymbol{x}_0 \mid \boldsymbol{x}_t, \hat{\boldsymbol{y}}_0^{(t)}, \boldsymbol{x}_{cor})$
9: $\hat{\boldsymbol{y}}_0^{(t)} \sim p_\theta^{(y)}(\boldsymbol{y}_0 \mid \boldsymbol{y}_t, \hat{\boldsymbol{x}}_0^{(t)}, \boldsymbol{x}_{cor})$
10: $\hat{\boldsymbol{\epsilon}}_x \leftarrow \text{Denoiser}_\theta^{(x)}(\boldsymbol{x}_t, \boldsymbol{y}_t, \hat{\boldsymbol{y}}_0^{(t)}, \boldsymbol{x}_{cor}, t)$
11: $\hat{\boldsymbol{\epsilon}}_y \leftarrow \text{Denoiser}_\theta^{(y)}(\boldsymbol{y}_t, \boldsymbol{x}_t, \hat{\boldsymbol{x}}_0^{(t)}, \boldsymbol{x}_{cor}, t)$
12: $\mathcal{L} \leftarrow \|\hat{\boldsymbol{\epsilon}}_x - \boldsymbol{\epsilon}_x\|_2^2 + \|\hat{\boldsymbol{\epsilon}}_y - \boldsymbol{\epsilon}_y\|_2^2$
13: Apply optimization step on $\mathcal{L}$ to update $\theta$
14: **end repeat**

### 3.5. Nested Strategy

Finally, the *Nested* strategy (Fig. 2(c)) adopts a *hierarchical scheduler*. It performs a single diffusion loop over the logits while embedding a complete signal diffusion process at each logit timestep. By ensuring that logit updates are conditioned on a high-quality estimate of the clean signal, this strategy avoids reliance on potentially inaccurate intermediate predictions, at the cost of increased computational complexity.

**Proposition 3.2.** *Given a corrupted observation $\boldsymbol{x}_{cor}$, the* Nested *strategy produces samples that approximately maximize the conditional logit trajectory probability*

$$p(\boldsymbol{y}_{0:T-1} \mid \boldsymbol{y}_T, \boldsymbol{x}_{cor}) = \prod_{t=1}^{T} p(\boldsymbol{y}_{t-1} \mid \boldsymbol{y}_t, \boldsymbol{x}_{cor}), \quad (16)$$

*where each transition $p(\boldsymbol{y}_{t-1} \mid \boldsymbol{y}_t, \boldsymbol{x}_{cor})$ is approximated by marginalizing over a clean signal estimate obtained from a full inner diffusion trajectory conditioned on $\boldsymbol{y}_t$.*

Further motivation, the inference algorithm, probabilistic derivations, and training details are provided in Appendix C.

**Practical compute reduction.** Executing a full inner signal diffusion loop at every logit timestep can be computationally expensive. In practice, we substantially reduce cost by running the inner signal diffusion once at the beginning to obtain an initial signal estimate, and updating it only during the later stages of the logit reverse process, where high-fidelity signal guidance is most critical. Specifically, for early reverse-time steps, $t > t_{\text{switch}}$, the signal estimate is kept fixed and no inner diffusion is performed. The inner signal diffusion is activated only for $t \leq t_{\text{switch}}$, corresponding to the final portion of the reverse trajectory.

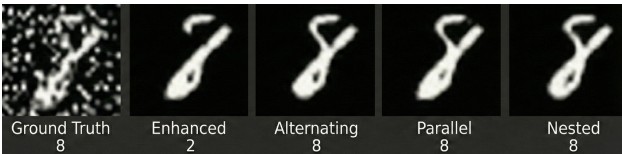

*Figure 3.* Qualitative comparison under 30% GN. The enhanced baseline incorrectly recognizes "2," whereas our coupled strategies (*Alternating*, *Parallel*, and *Nested*) correctly recover the digit by leveraging logit-space guidance, reconstructing the missing upper-left stroke, enabling correct recognition as "8."

Optionally, executing the inner loop every few outer steps further reduces computation while preserving much of the guidance quality.

## 4. Experiments

We evaluated our proposed framework on two modalities: image classification and ASR. We compared our three strategies against multiple baselines. The *Noisy* baseline evaluates a standard pre-trained classifier directly on corrupted inputs. The *Enhanced* baseline corresponds to a two-stage "enhance-then-classify" pipeline, in which an unconditional diffusion model first restores the input signal independently of the classifier, followed by downstream classification. Finally, the *CARD* baseline (Han et al., 2022) generates classifier logits conditioned on a fixed corrupted image. We also compare to enhancers trained with both regression and classification losses, namely *URIE* (Son et al., 2020) for images and *Dissen* (Dissen et al., 2025) for audio. These provide a strong comparison to methods that explicitly incorporate task supervision during enhancement.

### 4.1. Robust Image Classification

We first analyzed the performance of our coupled strategies on image classification tasks, which serve as a proof-of-concept before evaluating the framework in more realistic speech settings. We evaluated all the approaches on four datasets: CIFAR-10, CIFAR-100 (Krizhevsky, 2009), and the ImageNet dataset (Chrabaszcz et al., 2017). In addition, we included MNIST (LeCun et al., 2002) as a low-complexity grayscale benchmark, used primarily for qualitative illustration of the coupled denoising behavior. For ImageNet, we used the dataset at $32 \times 32$ resolution and a random 100-class subset, sampled once with a fixed random seed and shared across all experiments. It will be denoted as *ImageNet32-100* (IN32-100).

To simulate several environmental degradations, we introduced three distinct noise types during evaluation:

1. **15% White Noise (15% GN):** randomly replace 15% of the image pixels with standard white noise.

2. **30% White Noise (30% GN):** randomly replace 30% of the image pixels with standard white noise.

3. **Gaussian Blur (GB):** application of a Gaussian kernel of size $5 \times 5$ with a blur of $\sigma = 2$, simulating sensor defocus or motion blur.

Implementation details for the image experiments, including network architectures, training configuration, DDIM-based sampling, and sampling efficiency (in terms of NFE) across strategies, are provided in Appendix D.

The results are summarized in Table 1. Each row corresponds to a dataset under a specific noise level, and each column corresponds to a method. Across all non-trivial image datasets and corruption settings, the coupled strategies consistently outperform the baseline approaches. An exception is the simple MNIST dataset, where URIE achieves slightly higher accuracy, likely due to the low complexity of the data and the limited benefit of iterative coupling in such settings. The weaker performance of the *Noisy* and *CARD* baselines can be attributed to the fact that the classifier is trained only on clean images, and thus does not generalize well to corrupted inputs; moreover, in *CARD*, the logits denoiser conditions solely on the corrupted signal, which limits its ability to recover reliable semantic information. Among the proposed methods, *Alternating* and *Nested* achieve the strongest robustness on the CIFAR and MNIST datasets, while *Parallel* scales more favorably to more complex data, such as ImageNet32, highlighting a trade-off between coupling strength and optimization stability. Further discussion of the differences between the inference strategies is provided in Appendix E.

Qualitative example of enhancement behavior of each method on MNIST is shown in Fig. 3. The leftmost panel presents an example with 30% Gaussian noise on MNIST. The enhanced baseline incorrectly recognizes "2," whereas our coupled strategies (*Alternating*, *Parallel*, and *Nested*) correctly recover the digit by leveraging logit-space guidance, reconstructing the missing upper-left stroke, enabling correct recognition as "8." Additional qualitative examples are provided in Appendix D.5, where it can be observed that the generated digits are progressively shaped to be more easily classifiable.

## 4.2. Robust Speech Recognition

We further evaluated the proposed methods on the ASR task, where the temporal structure of speech signals introduces distinct modeling challenges. Our experiments employed Whisper-base (Radford et al., 2023), a state-of-the-art transformer-based ASR model. In this setting, the logits $\boldsymbol{y}_t$ correspond to a sequence of pre-softmax token logits, with one logit vector produced at each autoregressive decoding step of the Whisper decoder. This sequence-level representation provides semantic guidance that steers the

*Table 1.* Classification accuracy (%) on image datasets, with pre-trained classifier (Noisy), Enhanced signal baseline, URIE (enhancer trained with regression and classification losses), static conditional generation of logits (CARD), and our coupled strategies. Best robust results are highlighted in bold.

| Dataset | Corruption | Baselines (%) ↑ | | | | Coupled (Ours) (%) ↑ | | |
|---|---|---|---|---|---|---|---|---|
| | | Noisy | Enhanced | URIE | CARD | Parallel | Nested | Alternating |
| **MNIST** | 15% GN | 89.7 | 98.4 | **99.4** | 86.3 | 99.2 | 99.3 | 98.9 |
| | 30% GN | 66.7 | 98.3 | **99.3** | 66.2 | 99.2 | 99.1 | 98.7 |
| | Gaussian Blur | 89.6 | 98.3 | **99.5** | 87.0 | 99.2 | 99.3 | 98.6 |
| **CIFAR-10** | 15% GN | 16.6 | 60.9 | 68.5 | 11.3 | 74.3 | 81.3 | **83.0** |
| | 30% GN | 10.9 | 60.4 | 59.4 | 10.4 | 71.6 | 81.2 | **82.8** |
| | Gaussian Blur | 34.8 | 60.1 | 66.9 | 14.5 | 80.9 | 80.8 | **83.3** |
| **CIFAR-100** | 15% GN | 6.7 | 61.2 | 68.2 | 8.5 | 70.7 | **74.1** | 70.5 |
| | 30% GN | 2.0 | 60.4 | 54.08 | 6.1 | 68.5 | **71.9** | 68.6 |
| | Gaussian Blur | 10.4 | 60.4 | 64.6 | 10.2 | 64.8 | **73.8** | 69.7 |
| **IN32-100** | 15% GN | 1.8 | 52.9 | 51.1 | 3.0 | **69.8** | 67.5 | 64.8 |
| | 30% GN | 1.0 | 50.6 | 26.8 | 1.2 | **65.5** | 62.0 | 58.1 |
| | Gaussian Blur | 2.5 | 51.3 | 33.6 | 5.9 | 63.3 | **65.8** | 62.5 |

signal diffusion process toward ASR-intelligible speech.

For speech enhancement, we used the Score-Based Generative Model for Speech Enhancement (SGMSE) (Richter et al., 2023), which formulates speech enhancement as a conditional score-based diffusion process in the complex short-time Fourier transform (STFT) domain. Comprehensive implementation details, including the SGMSE formulation, integration with Whisper, network architectures, logit handling, inference settings, and a comparison of sampling efficiency (in terms of NFE) across different strategies, are provided in Appendix F.

We evaluated all methods on three datasets. The Google Speech Commands dataset (Warden, 2018) consists of isolated utterances of 10 words spoken by more than 2,000 speakers. We augmented this dataset with artificial reverberation. The EARS dataset (Richter et al., 2024) comprises over 100 hours of clean, anechoic speech. To assess robustness, we evaluated EARS under two conditions. The first, denoted *EARS-Reverb*, is constructed by convolving the clean speech signals with simulated reverberant room impulse responses. The second, denoted *EARS-WHAM*, is constructed by adding noise sampled from the WHAM noise corpus (Wichern et al., 2019). Additional details on dataset preprocessing and experimental conditions are given in Appendix F.2.

Table 2 presents the Word Error Rate (WER) across different methods, where lower is better. Each row represents a different dataset. We can see that coupling signal enhancement with evolving semantic logits consistently improves WER compared to the baselines. Specifically, the *Nested* strategy consistently yields the lowest WER across all datasets, with coupled methods exhibiting larger gains as the acoustic complexity or task difficulty increases. By conditioning the enhancement process on intermediate Whisper logits, the model is guided toward speech representations that remain aligned with the downstream recognizer, improving transcription reliability under distortions.

*Table 2.* WER (%) comparison on Google Commands, EARS-Reverb and EARS-WHAM. Noisy denotes the performance of Whisper-base. Enhanced denotes audio-only enhancement, Dissen denotes an enhancer trained with regression and classification losses, and CARD denotes logits-only enhancement.

| | Baselines (%) ↓ | | | | Coupled (Ours) (%) ↓ | | |
|---|---|---|---|---|---|---|---|
| Dataset | Noisy | Enhanced | Dissen | CARD | Parallel | Nested | Alternating |
| **G.Cmds-Reverb** | 5.58 | 5.11 | 7.96 | 11.08 | 4.98 | **4.85** | 5.03 |
| **EARS-Reverb** | 5.79 | 4.48 | 6.46 | 16.79 | 3.95 | **3.83** | 4.25 |
| **EARS-WHAM** | 14.02 | 10.04 | 13.04 | 36.58 | 9.90 | **9.75** | 10.16 |

## 4.3. Computational Complexity.

We conclude by briefly discussing the computational cost of the proposed strategies. Measured in terms of neural function evaluations (NFE), the *Parallel*, *Alternating*, and *Nested* strategies incur approximately $2\times$, $10\times$, and $7\times$ the cost of a standard single-modality diffusion model, respectively. However, these ratios do not directly translate to wall-clock runtime. A key observation is that logits diffusion is often computationally cheaper than signal diffusion, as it operates in a lower-dimensional space. In our experiments on the EARS dataset, a single signal NFE was approximately $7\times$ more expensive than a logits NFE. As a result, the additional cost introduced by logits diffusion is relatively minor, and the overall runtime increase, particularly for the *Parallel* strategy, is considerably smaller than what NFE alone would suggest.

Beyond this, the proposed framework is agnostic to the underlying generative backbone, and thus readily compatible with standard acceleration techniques such as improved ODE/SDE solvers (Lu et al., 2022; Zheng et al., 2023) or model distillation (Salimans & Ho, 2022) to reduce the number of sampling steps. Additional orthogonal directions for improving efficiency include temporal feature caching, such as DeepCache (Ma et al., 2024), to reduce per-step computation, as well as adopting Flow Matching (Lipman et al., 2022) or Rectified Flow (Liu et al., 2022b) formulations to enable faster convergence. In the context of ASR, operating in a compressed latent space via neural audio codecs (Défossez et al., 2022; Zeghidour et al., 2021) offers a promising avenue for further reducing computational overhead.

A more detailed analysis of computational cost, including modality-specific breakdowns, is provided in Appendix D.4 for image experiments and Appendix F.5 for audio experiments.

## 4.4. Ablation Studies

We conclude the experiments by considering different aspect of the proposed approach using several ablation studies.

**Effect of Sampling Steps.** We examined the sensitivity of our approach to the number of diffusion sampling steps $T$. We executed our three strategies on varying diffusion steps from 1 to 150. Results are shown in Fig. 4, evaluating CIFAR-100 and ImageNet32-100 under 30% Gaussian noise. On both datasets, the *Parallel* strategy appears to exhibit rapid convergence, reaching stable performance within very few steps ($< 10$). In contrast, the *Nested* and *Alternating* methods generally require more steps to saturate. The strong performance of the *Parallel* strategy stems from its tight, step-level coupling: at each denoising step, both processes are immediately informed by the latest state of the other, enabling continuous mutual refinement. This feedback rapidly aligns signal reconstruction with evolving semantic predictions, allowing errors to be corrected early. As a result, high-quality predictions are achieved in significantly fewer steps. This behavior is consistent across datasets, suggesting it is driven by the coupling mechanism rather than dataset-specific factors. These trends are consistent across datasets, suggesting that the convergence behavior is driven by the coupling strategy rather than dataset-specific factors. We plan to explore this in future work.

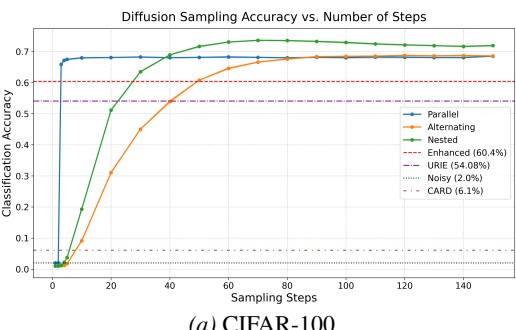

*(a)* CIFAR-100

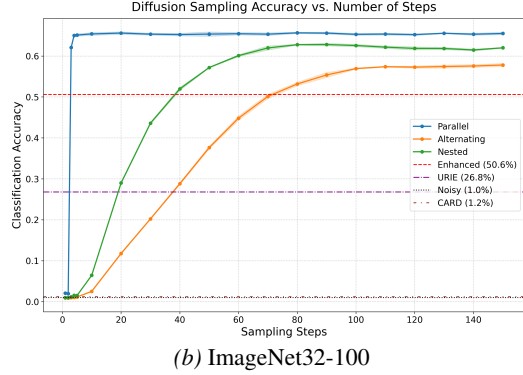

*(b)* ImageNet32-100

*Figure 4.* Robustness vs. Sampling Steps. Accuracy as a function of the number of diffusion steps on CIFAR-100 and ImageNet32-100 under 30% GN. The *Parallel* strategy converges rapidly, while *Nested* and *Alternating* require more steps to saturate.

**Guidance Source in Parallel Sampling.** In this part we would like to consider whether to guide the update using the current noisy sample (i.e., $x_{t-1}$ or $y_t$) or the

corresponding clean estimate (i.e., $\hat{\boldsymbol{x}}_0^{(t)}$ or $\hat{\boldsymbol{y}}_0^{(t+1)}$). This is mostly relevant for the *Parallel* strategy, where updating one modality at timestep $t$ requires conditioning on the other modality at an intermediate (partially denoised) timestep. The noisy sample reflects a conservative update that remains closely tied to the diffusion posterior, while the clean estimate provides a more semantically informative signal but relies more heavily on the model's prediction of the previous step.

To study this trade-off, we compared these two guidance options within the *Parallel* strategy on CIFAR-100 and ImageNet32-100 across all corruption types. Results are reported in Table 3. It can be seen that conditioning on the clean estimate consistently yields higher classification accuracy than conditioning on the noisy sample. This indicates that, in the coupled setting, the semantic clarity of the clean estimate outweighs the increased dependence on the predictive model, and we therefore adopt clean-estimate guidance in all experiments.

*Table 3.* Effect of guidance source in the *Parallel* strategy. We compared conditioning on the clean estimate ($\hat{\boldsymbol{x}}_0^{(t)}$, $\hat{\boldsymbol{y}}_0^{(t+1)}$) versus the noisy sample ($\boldsymbol{x}_{t-1}$, $\boldsymbol{y}_t$). Classification accuracy (%) is reported.

| Dataset | Corruption | Clean-estimate guidance | Noisy-sample guidance |
|---------|-----------|------------------------|----------------------|
| **CIFAR-100** | 15% GN | **70.7** | 70.0 |
| | 30% GN | **68.5** | 67.0 |
| | Gaussian Blur | **64.8** | 64.0 |
| **IN32-100** | 15% GN | **69.8** | 69.6 |
| | 30% GN | **65.5** | 64.6 |
| | Gaussian Blur | **63.3** | 61.8 |

**Sampling Algorithm: DDPM vs. DDIM.** In the last ablation, we compared stochastic DDPM sampling with deterministic DDIM for the *Parallel* strategy on CIFAR-100. As shown in Table 4, DDPM consistently outperforms DDIM across all corruption types. The performance gap is modest under Gaussian noise but becomes more pronounced under Gaussian blur, suggesting that stochastic sampling better mitigates corruption by avoiding premature convergence to suboptimal trajectories.

*Table 4.* Evaluation of DDPM versus DDIM sampling algorithms for the *Parallel* strategy on CIFAR-100. Classification accuracy (%) is reported.

| Corruption | DDPM | DDIM |
|-----------|------|------|
| 15% GN | **73.5** | 70.7 |
| 30% GN | **71.2** | 68.5 |
| Gaussian Blur | **69.7** | 64.8 |

**Scaling to Standard Image Resolution.** While our main experiments focus on low-resolution benchmarks (up to $32 \times 32$) due to computational resource constraints, we also

evaluate the scalability of the proposed framework to standard image resolutions. Specifically, we conduct a proof-of-concept experiment on ImageNet at $224 \times 224$ resolution. Due to the dataset scale, we restrict this experiment to a 10-class subset, following a fixed sampling protocol.

In this setting, we compare our *Parallel* strategy against the *Enhanced* baseline, which was the strongest baseline on more complex datasets such as ImageNet32-100. The goal of this experiment is not to provide a comprehensive benchmark, but rather to assess whether the proposed mutual guidance mechanism remains stable and effective at higher resolutions.

Table 5 shows the results under different corruption types. We observe that the proposed method consistently improves over the *Enhanced* baseline across all settings, with the gains becoming more pronounced as the corruption severity increases. In particular, under stronger noise (e.g., 30% GN), where the task is more challenging, the advantage of the coupled approach is significantly larger. These results suggest that the coupled diffusion framework can scale to standard-resolution images, with the observed computational overhead being consistent with that of diffusion-based methods in general, rather than introducing new bottlenecks.

*Table 5.* Scaling to full-resolution ImageNet ($224 \times 224$, 10 classes). Classification accuracy (%) under different corruptions. We compare the *Enhanced* baseline with our *Parallel* strategy.

| Corruption | Enhanced | Parallel (Ours) |
|-----------|----------|-----------------|
| 15% GN | 92.1 | **94.0** |
| 30% GN | 86.5 | **92.4** |
| Gaussian Blur | 94.0 | **94.1** |

## 5. Conclusion

We introduced a general framework for robust classification based on coupled diffusion processes over signals and logits, enabling mutual guidance between signal enhancement and class prediction refinement without modifying the underlying classifier. Across both image classification and speech recognition, our approach consistently improves robustness under diverse corruptions, outperforming sequential enhancement, static generative baselines, and enhancement methods that incorporate task supervision (e.g., URIE and Dissen) in most settings. The proposed *Parallel*, *Alternating*, and *Nested* strategies offer flexible trade-offs between computational cost and coupling strength. These findings point toward a broader paradigm in which generative models are not merely used for data synthesis or enhancement, but serve as adaptive inference engines that reshape inputs in service of downstream decision-making.

## Impact Statement

This paper presents work whose goal is to advance the field of Machine Learning. There are many potential societal consequences of our work, none which we feel must be specifically highlighted here.

## Acknowledgements

We acknowledge the support this project received through the ATS Fund for Applied Security Science and Technology Research.

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

# A. On the Redundancy of Conditioning Variables in the Reverse Posterior

For simplicity, we present the argument for the signal diffusion process; the same reasoning applies analogously to the logit diffusion. Recall that in the standard DDPM formulation, Eq. (3), the reverse posterior $q(\boldsymbol{x}_{t-1} \mid \boldsymbol{x}_t, \hat{\boldsymbol{x}}_0^{(t)})$ depends only on the current noisy state $\boldsymbol{x}_t$ and an estimate of the clean sample $\hat{\boldsymbol{x}}_0^{(t)}$. In practice, $\hat{\boldsymbol{x}}_0^{(t)}$ is not predicted directly, but is obtained from a neural denoiser that estimates the additive diffusion noise, from which $\hat{\boldsymbol{x}}_0^{(t)}$ is recovered using the diffusion reparameterization.

Additional variables such as $\boldsymbol{y}_t$ and $\boldsymbol{x}_{cor}$ are introduced only as conditioning inputs to this denoising network. Once the estimate $\hat{\boldsymbol{x}}_0^{(t)}$ is fixed, these variables no longer affect the Gaussian posterior itself and are therefore redundant in the conditioning of $q$. They serve purely as auxiliary guidance to the network that produces the noise estimate used to recover $\hat{\boldsymbol{x}}_0^{(t)}$, but do not alter the analytic form of the reverse transition.

An important exception arises in the audio experiments, where we instantiate the signal diffusion using the SGMSE framework (Richter et al., 2023) (Appendix F.1). In this setting, the forward process does not diffuse the clean signal directly toward Gaussian noise, but instead interpolates between the clean signal and the corrupted observation $\boldsymbol{x}_{cor}$. Specifically, the forward SDE induces a mean trajectory $\boldsymbol{\mu}(t)$ that is a time-dependent interpolation between $\boldsymbol{x}_0$ and $\boldsymbol{x}_{cor}$. Consequently, the reverse-time posterior effectively takes the form $q(\boldsymbol{x}_{t-1} \mid \boldsymbol{x}_t, \boldsymbol{\mu}(t))$, where $\boldsymbol{\mu}(t)$ depends explicitly on $\boldsymbol{x}_{cor}$.

Under this formulation, each reverse step can be interpreted as an interpolation between $\boldsymbol{x}_t$ and $\boldsymbol{\mu}(t)$, rather than between $\boldsymbol{x}_t$ and $\hat{\boldsymbol{x}}_0^{(t)}$. Since $\boldsymbol{\mu}(t)$ is a function of $\boldsymbol{x}_{cor}$, the corrupted signal remains a necessary conditioning variable in the posterior for the SGMSE-based audio models. This explains why, in the audio experiments, $\boldsymbol{x}_{cor}$ cannot be dropped from the conditioning of $q$, whereas in the standard DDPM setting used for images, it is formally redundant.

For simplicity of presentation, we omit $\boldsymbol{x}_{cor}$ from the conditioning in the algorithms throughout the paper and write the reverse transitions in their reduced form. In practice, however, for the audio experiments based on SGMSE, the corrupted signal $\boldsymbol{x}_{cor}$ is explicitly used in the diffusion process, as discussed above.

# B. Alternating Strategy: Additional Details

## B.1. Motivation and Design Rationale

The Alternating strategy is motivated by the observation that, while tightly interleaved coupling (as in the Parallel strategy) enables maximal information exchange between signal and logits, it also conditions each update on intermediate diffusion states that may be noisy, unstable, or semantically unreliable, particularly at early timesteps. Conditioning on such intermediate states can propagate noise across modalities and complicate the probabilistic structure of the reverse process.

To address this, the Alternating strategy enforces a temporal separation between signal and logit refinement. Instead of interleaving updates at each diffusion step, each modality is updated only after the other has completed a full reverse diffusion trajectory and reached a fully denoised point estimate. These terminal estimates, $\hat{\boldsymbol{x}}_0^i$ and $\hat{\boldsymbol{y}}_0^i$, are treated as deterministic and semantically stable conditioning variables in subsequent iterations.

Importantly, when the number of alternating iterations is set to $N = 1$, the procedure reduces to a diffusion-based *enhance-then-classify* pipeline: a single signal denoising trajectory produces $\hat{\boldsymbol{x}}_0^1$, which is then used to generate logits in a subsequent diffusion pass. Increasing $N$ generalizes this paradigm to an iterative joint refinement scheme, enabling repeated feedback between signal enhancement and classification.

## B.2. Alternating Inference Procedure

We provide here the explicit inference procedure for the *Alternating* strategy, corresponding to the formulation in Sec. 3.4. The method proceeds in outer iterations, where each iteration consists of a full reverse diffusion trajectory over the signal, conditioned on the current logit estimate, followed by a full reverse diffusion trajectory over the logits, conditioned on the newly obtained signal estimate. The clean point estimates $\hat{\boldsymbol{x}}_0^i$ and $\hat{\boldsymbol{y}}_0^i$ are treated as deterministic conditioning variables across iterations.

Algorithm 4 summarizes the complete inference process.

## B.3. Probabilistic Derivation

We derive the probabilistic objective underlying the *Alternating* strategy and show how Algo. 4 follows from a structured approximation of the joint reverse-time distribution.

**Joint reverse-time objective** As in the Parallel strategy, at inference time we aim to sample trajectories from the joint reverse-time distribution

$$p(\boldsymbol{x}_{0:T-1}, \boldsymbol{y}_{0:T-1} \mid \boldsymbol{x}_T, \boldsymbol{y}_T, \boldsymbol{x}_{cor}),$$

where $\boldsymbol{x}_T \sim \mathcal{N}(0, I)$ and $\boldsymbol{y}_T \sim \mathcal{N}(0, I)$.

In contrast to the Parallel strategy, the Alternating method does not interleave signal and logit updates at each reverse-time step. Instead, it performs complete diffusion trajectories for the signal and logits sequentially, conditioning each

**Algorithm 4** Alternating Inference Algorithm

---

1: $\hat{\boldsymbol{y}}_0^0 \leftarrow f_\phi(\boldsymbol{x}_{cor})$
2: **for** $i = 1$ **to** $N$ **do**
3:     *// Signal Diffusion Trajectory*
4:     $\boldsymbol{x}_T \sim \mathcal{N}(0, \boldsymbol{I})$
5:     **for** $t = T$ **to** 1 **do**
6:         $\hat{\boldsymbol{x}}_0^{(t)} \sim p_\theta^{(x)}\big(\boldsymbol{x}_0 \mid \boldsymbol{x}_t^i, \hat{\boldsymbol{y}}_0^{i-1}, \boldsymbol{x}_{cor}\big)$
7:         $\boldsymbol{x}_{t-1}^i \sim q(\boldsymbol{x}_{t-1}^i \mid \boldsymbol{x}_t^i, \hat{\boldsymbol{x}}_0^{(t)})$
8:     **end for**
9:     $\hat{\boldsymbol{x}}_0^i \leftarrow \boldsymbol{x}_0^i$
10:     *// Logit Diffusion Trajectory*
11:     $\boldsymbol{y}_T \sim \mathcal{N}(0, \boldsymbol{I})$
12:     **for** $t = T$ **to** 1 **do**
13:         $\hat{\boldsymbol{y}}_0^{(t)} \sim p_\theta^{(y)}\big(\boldsymbol{y}_0 \mid \boldsymbol{y}_t^i, \hat{\boldsymbol{x}}_0^i, \boldsymbol{x}_{cor}\big)$
14:         $\boldsymbol{y}_{t-1}^i \sim q(\boldsymbol{y}_{t-1}^i \mid \boldsymbol{y}_t^i, \hat{\boldsymbol{y}}_0^{(t)})$
15:     **end for**
16:     $\hat{\boldsymbol{y}}_0^i \leftarrow \boldsymbol{y}_0^i$
17: **end for**
18: **return** $\hat{\boldsymbol{y}}_0^N$

---

process on a fixed estimate obtained from the previous iteration.

**Factorization via conditional independence** Applying the chain rule yields

$$p(\boldsymbol{x}_{0:T-1}, \boldsymbol{y}_{0:T-1} \mid \boldsymbol{x}_T, \boldsymbol{y}_T, \boldsymbol{x}_{cor}) =$$
$$p(\boldsymbol{x}_{0:T-1} \mid \boldsymbol{x}_T, \boldsymbol{y}_T, \boldsymbol{x}_{cor}) \cdot$$
$$p(\boldsymbol{y}_{0:T-1} \mid \boldsymbol{x}_{0:T}, \boldsymbol{y}_T, \boldsymbol{x}_{cor}). \quad (17)$$

We exploit the following conditional independence properties of the diffusion processes:

- **Signal independence from logit noise.** Given $\boldsymbol{x}_T$ and the corrupted observation $\boldsymbol{x}_{cor}$, the signal trajectory $\boldsymbol{x}_{0:T-1}$ is independent of the initial logit noise $\boldsymbol{y}_T$.
- **Logit independence from intermediate signal states.** Given the clean signal $\boldsymbol{x}_0$, the logit trajectory $\boldsymbol{y}_{0:T-1}$ is independent of the intermediate noisy signal states $\boldsymbol{x}_{1:T}$.

Under these assumptions in addition to the Markovian assumption, the joint distribution simplifies to

$$p(\boldsymbol{x}_{0:T-1}, \boldsymbol{y}_{0:T-1} \mid \boldsymbol{x}_T, \boldsymbol{y}_T, \boldsymbol{x}_{cor})$$
$$= p(\boldsymbol{x}_{0:T-1} \mid \boldsymbol{x}_T, \boldsymbol{x}_{cor}) \, p(\boldsymbol{y}_{0:T-1} \mid \boldsymbol{x}_0, \boldsymbol{y}_T, \boldsymbol{x}_{cor})$$
$$= \prod_{t=1}^{T} p(\boldsymbol{x}_{t-1} \mid \boldsymbol{x}_t, \boldsymbol{x}_{cor}) \prod_{t=1}^{T} p(\boldsymbol{y}_{t-1} \mid \boldsymbol{y}_t, \boldsymbol{x}_0, \boldsymbol{x}_{cor}).$$
$$(18)$$

**Iterative conditioning and approximation** We now derive tractable approximations for the signal and logit transitions in the Alternating strategy, following the same

reverse-time factorization and point-estimate approximations used in Section 3.3. In contrast to the Parallel method, the Alternating strategy executes a full reverse trajectory for $\boldsymbol{x}$ using a *fixed* logit estimate from the previous outer iteration, and then executes a full reverse trajectory for $\boldsymbol{y}$ using the resulting *fixed* signal estimate. We therefore derive the per-timestep update rules while keeping the outer iteration index implicit, and introduce $\hat{\boldsymbol{y}}_0^{i-1}$ only at the point where it enters the denoiser.

**Signal update.** Consider the signal transition distribution $p(\boldsymbol{x}_{t-1} \mid \boldsymbol{x}_t, \boldsymbol{x}_{cor})$. In the Alternating strategy, the reverse-time sampling is guided by an auxiliary clean-logit estimate from the previous outer iteration, denoted $\hat{\boldsymbol{y}}_0^{i-1}$, which is held fixed for the entire signal trajectory. This guidance is used to improve the prediction of the clean signal $\hat{\boldsymbol{x}}_0^{(t)}$.

As in the Parallel derivation, we introduce the clean signal $\boldsymbol{x}_0$ and marginalize:

$$p(\boldsymbol{x}_{t-1} \mid \boldsymbol{x}_t, \boldsymbol{x}_{cor}) =$$
$$\int p(\boldsymbol{x}_{t-1} \mid \boldsymbol{x}_t, \boldsymbol{x}_0, \boldsymbol{x}_{cor}) \, p(\boldsymbol{x}_0 \mid \boldsymbol{x}_t, \boldsymbol{x}_{cor}) \, d\boldsymbol{x}_0. \quad (19)$$

This integral is intractable due to the unknown posterior $p(\boldsymbol{x}_0 \mid \boldsymbol{x}_t, \boldsymbol{x}_{cor})$. Following standard diffusion practice (Ho et al., 2020), we approximate this posterior with a point-mass at a denoised estimate $\hat{\boldsymbol{x}}_0^{(t)}$. In the Alternating strategy, this point estimate is predicted by a conditional denoiser that incorporates the fixed logit estimate $\hat{\boldsymbol{y}}_0^{i-1}$:

$$p(\boldsymbol{x}_0 \mid \boldsymbol{x}_t, \boldsymbol{x}_{cor}) \approx \delta(\boldsymbol{x}_0 - \hat{\boldsymbol{x}}_0^{(t)}),$$
$$\hat{\boldsymbol{x}}_0^{(t)} \sim p_\theta^{(x)}(\boldsymbol{x}_0 \mid \boldsymbol{x}_t, \hat{\boldsymbol{y}}_0^{i-1}, \boldsymbol{x}_{cor}). \quad (20)$$

Substituting (20) into (19) collapses the integral and yields

$$p(\boldsymbol{x}_{t-1} \mid \boldsymbol{x}_t, \boldsymbol{x}_{cor}) \approx q\Big(\boldsymbol{x}_{t-1} \mid \boldsymbol{x}_t, \hat{\boldsymbol{x}}_0^{(t)}, \boldsymbol{x}_{cor}\Big), \quad (21)$$

where $q(\cdot)$ denotes the tractable Gaussian posterior induced by the forward diffusion process. Importantly, conditioned on $(\boldsymbol{x}_t, \hat{\boldsymbol{x}}_0^{(t)})$, the posterior is independent of $\boldsymbol{x}_{cor}$, which only affects the transition through the estimate $\hat{\boldsymbol{x}}_0^{(t)}$:

$$q(\boldsymbol{x}_{t-1} \mid \boldsymbol{x}_t, \hat{\boldsymbol{x}}_0^{(t)}, \boldsymbol{x}_{cor}) = q(\boldsymbol{x}_{t-1} \mid \boldsymbol{x}_t, \hat{\boldsymbol{x}}_0^{(t)}),$$

See Appendix A for details. Hence, each reverse-time step for the signal consists of (i) predicting $\hat{\boldsymbol{x}}_0^{(t)}$ using $p_\theta^{(x)}(\cdot \mid \boldsymbol{x}_t, \hat{\boldsymbol{y}}_0^{i-1}, \boldsymbol{x}_{cor})$, and (ii) sampling $\boldsymbol{x}_{t-1}$ from the corresponding posterior $q(\boldsymbol{x}_{t-1} \mid \boldsymbol{x}_t, \hat{\boldsymbol{x}}_0^{(t)})$.

**Logit update.** We now derive the corresponding approximation for the logit transition $p(\boldsymbol{y}_{t-1} \mid \boldsymbol{y}_t, \boldsymbol{x}_0, \boldsymbol{x}_{cor})$. Following the same procedure as in the Parallel strategy, we introduce the clean logits $\boldsymbol{y}_0$, and marginalize:

$$p(\boldsymbol{y}_{t-1} \mid \boldsymbol{y}_t, \boldsymbol{x}_0, \boldsymbol{x}_{cor}) =$$
$$\int p(\boldsymbol{y}_{t-1} \mid \boldsymbol{y}_t, \boldsymbol{y}_0, \boldsymbol{x}_0, \boldsymbol{x}_{cor}) \cdot p(\boldsymbol{y}_0 \mid \boldsymbol{y}_t, \boldsymbol{x}_0, \boldsymbol{x}_{cor}) \, d\boldsymbol{y}_0. \quad (22)$$

After completing the signal diffusion trajectory, the Alternating strategy provides a fixed estimate of the clean signal, denoted $\hat{\boldsymbol{x}}_0^i$. We therefore approximate the dependence on the unknown clean signal using a point estimate,

$$p(\boldsymbol{y}_0 \mid \boldsymbol{y}_t, \boldsymbol{x}_0, \boldsymbol{x}_{cor}) \approx \delta(\boldsymbol{y}_0 - \hat{\boldsymbol{y}}_0^{(t)}),$$
$$\hat{\boldsymbol{y}}_0^{(t)} \sim p_\theta^{(y)}(\boldsymbol{y}_0 \mid \boldsymbol{y}_t, \hat{\boldsymbol{x}}_0^i, \boldsymbol{x}_{cor}). \quad (23)$$

Here, the fixed signal estimate $\hat{\boldsymbol{x}}_0^i$ enters through the prediction of the clean logits $\hat{\boldsymbol{y}}_0^{(t)}$ and remains constant throughout the entire logit diffusion trajectory.

Substituting (23) into (22) collapses the integral, yielding the tractable sampling distribution

$$p(\boldsymbol{y}_{t-1} \mid \boldsymbol{y}_t, \boldsymbol{x}_0, \boldsymbol{x}_{cor})$$
$$\approx q\Big(\boldsymbol{y}_{t-1} \mid \boldsymbol{y}_t, \hat{\boldsymbol{y}}_0^{(t)}, \boldsymbol{x}_{cor}\Big) = q\Big(\boldsymbol{y}_{t-1} \mid \boldsymbol{y}_t, \hat{\boldsymbol{y}}_0^{(t)}\Big), \quad (24)$$

where the last equality follows as in the signal update (see Appendix A). Consequently, each reverse-time step for the logit diffusion consists of (i) predicting $\hat{\boldsymbol{y}}_0^{(t)}$ conditioned on the fixed signal estimate $\hat{\boldsymbol{x}}_0^i$, and (ii) sampling $\boldsymbol{y}_{t-1}$ from the corresponding posterior distribution.

**Resulting joint reverse-time approximation** Combining the reverse-time factorization in (18) with the point-estimate approximations in Section B.3, the Alternating strategy yields the following approximation to the joint reverse-time distribution:

$$p(\boldsymbol{x}_{0:T-1}, \boldsymbol{y}_{0:T-1} \mid \boldsymbol{x}_T, \boldsymbol{y}_T, \boldsymbol{x}_{cor})$$
$$\approx \prod_{t=1}^{T} p\big(\boldsymbol{x}_{t-1} \mid \boldsymbol{x}_t, \hat{\boldsymbol{y}}_0^{i-1}, \boldsymbol{x}_{cor}\big) \cdot$$
$$\prod_{t=1}^{T} p\big(\boldsymbol{y}_{t-1} \mid \boldsymbol{y}_t, \hat{\boldsymbol{x}}_0^i, \boldsymbol{x}_{cor}\big). \quad (25)$$

**Connection to Algo. 4.** This factorization directly induces the two-stage inference procedure in Algo. 4, with a signal reverse trajectory conditioned on $\hat{\boldsymbol{y}}_0^{i-1}$ followed by a logit reverse trajectory conditioned on $\hat{\boldsymbol{x}}_0^i$.

### B.4. Training Procedure

Training of the Alternating strategy follows the same joint noise-prediction objective used throughout this work, with the key distinction that cross-modal conditioning is performed using *fully denoised* point estimates rather than intermediate diffusion states.

Given a clean sample $\boldsymbol{x}_0$ and its corrupted observation $\boldsymbol{x}_{cor}$, we first obtain a clean logit target $\boldsymbol{y}_0 = f_\phi(\boldsymbol{x}_0)$. To mimic the inference-time conditioning structure, we then generate point estimates $\hat{\boldsymbol{x}}_0$ and $\hat{\boldsymbol{y}}_0$ by running full reverse diffusion trajectories conditioned on $\boldsymbol{x}_{cor}$ and $\hat{\boldsymbol{x}}_0$, respectively.

At each optimization step, a diffusion timestep $t$ is sampled uniformly, Gaussian noise is injected to produce $(\boldsymbol{x}_t, \boldsymbol{y}_t)$, and separate denoisers are trained to predict the corresponding noise terms conditioned on the opposite modality's point estimate. The training loss is defined as the sum of signal and logit noise-prediction errors. This procedure explicitly aligns the training distribution with the alternating inference dynamics and enables stable cross-modal guidance.

The complete training algorithm is summarized in Algo. 5.

---

**Algorithm 5** Alternating Training Algorithm

---

1: **repeat** until convergence
2:   Sample $(\boldsymbol{x}_0, \boldsymbol{x}_{cor})$ from $\mathcal{D}$
3:   $\boldsymbol{y}_0 \leftarrow f_\phi(\boldsymbol{x}_0)$
4:   *// Full sampling to get estimates*
5:   $\hat{\boldsymbol{x}}_0 \leftarrow$ Full diffusion sampling conditioned on $\boldsymbol{x}_{cor}$
6:   $\hat{\boldsymbol{y}}_0 \leftarrow$ Full diffusion sampling conditioned on $\hat{\boldsymbol{x}}_0$
7:   *// Training iterations*
8:   **for** $k = 1$ **to** $K$ **do**
9:     Sample $t \sim \text{Uniform}(\{1, \dots, T\})$
10:     Sample $\boldsymbol{\epsilon}_x, \boldsymbol{\epsilon}_y \sim \mathcal{N}(0, \boldsymbol{I})$
11:     $\boldsymbol{x}_t \sim q(\boldsymbol{x}_t \mid \boldsymbol{x}_0, \boldsymbol{\epsilon}_x)$, $\boldsymbol{y}_t \sim q(\boldsymbol{y}_t \mid \boldsymbol{y}_0, \boldsymbol{\epsilon}_y)$
12:     $\hat{\boldsymbol{\epsilon}}_x \leftarrow \text{Denoiser}_\theta^{(\boldsymbol{x})}(\boldsymbol{x}_t, \hat{\boldsymbol{y}}_0, \boldsymbol{x}_{cor}, t)$
13:     $\hat{\boldsymbol{\epsilon}}_y \leftarrow \text{Denoiser}_\theta^{(y)}(\boldsymbol{y}_t, \hat{\boldsymbol{x}}_0, \boldsymbol{x}_{cor}, t)$
14:     $\mathcal{L} \leftarrow \|\hat{\boldsymbol{\epsilon}}_x - \boldsymbol{\epsilon}_x\|_2^2 + \|\hat{\boldsymbol{\epsilon}}_y - \boldsymbol{\epsilon}_y\|_2^2$
15:     Apply optimization step on $\mathcal{L}$ to update $\theta$
16:   **end for**
17: **end repeat**

---

## C. Nested Strategy: Additional Details

### C.1. Motivation and Design Rationale

A central challenge in coupled diffusion frameworks is the reliability of the conditioning information. In the *Parallel* strategy, the logit update at time $t$ relies on $\hat{\boldsymbol{x}}_0^{(t)}$, a point estimate of the clean signal derived from a noisy intermediate state $\boldsymbol{x}_t$. Especially during the early stages of the reverse process (large $t$), these estimates can be hallucinated or semantically unstable, potentially leading to cascading errors in the logit trajectory.

The Nested strategy addresses this limitation by ensuring that every transition in logit space is guided by a high-fidelity signal estimate. Inspired by the *Nested Diffusion*

framework of Elata et al. (2024), we treat a complete inner diffusion loop as the generative abstraction for signal reconstruction. Rather than relying on a single neural function evaluation, a full generative trajectory is executed to sample a clean signal estimate $\hat{x}_0$ conditioned on the current noisy logit state $y_t$. As shown in Elata et al. (2024), multi-step inner generative processes yield signal estimates that align more closely with the data manifold than single-shot predictions. This design ensures that each transition from $y_t$ to $y_{t-1}$ is informed by a semantically coherent signal reconstruction, improving the reliability of classification guidance at the cost of increased computational depth.

**Label-Centric Conditional Maximization.** Unlike the Parallel and Alternating strategies, which explicitly model joint signal-logit trajectories, the Nested approach can be interpreted as directly optimizing the conditional logit trajectory given the corrupted observation. The signal variable is introduced only through marginalization, serving as a latent bridge that improves the quality of the logit transitions without explicitly maintaining a coupled trajectory.

## C.2. Nested Inference Procedure

We provide here the explicit inference procedure for the *Nested* strategy, corresponding to the formulation in Sec. 3.5. The method consists of a single outer reverse diffusion process over the logits, where each reverse-time step is conditioned on a clean signal estimate obtained from a full inner signal diffusion trajectory. This implements the marginalization described in Proposition 3.2 by approximating each logit transition using a high-quality sample of the underlying clean signal.

Algorithm 6 summarizes the complete nested inference process.

---

**Algorithm 6** Nested Inference Algorithm

---

1: $y_T \sim \mathcal{N}(0, \boldsymbol{I})$
2: $\hat{y}_0^{(T+1)} \leftarrow f_\phi(\boldsymbol{x}_{cor})$
3: **for** $t = T$ **to** 1 **do**
4:    *// Inner signal diffusion loop*
5:    $\boldsymbol{x}_T \sim \mathcal{N}(0, \boldsymbol{I})$
6:    **for** $\tau = T$ **to** 1 **do**
7:       $\hat{\boldsymbol{x}}_0^{(\tau)} \sim p_\theta^{(x)}\Big(\boldsymbol{x}_0 \mid \boldsymbol{x}_\tau, \hat{y}_0^{(t+1)}, y_t, \boldsymbol{x}_{cor}\Big)$
8:       $\boldsymbol{x}_{\tau-1} \sim q(\boldsymbol{x}_{\tau-1} \mid \boldsymbol{x}_\tau, \hat{\boldsymbol{x}}_0^{(\tau)})$
9:    **end for**
10:    $\hat{\boldsymbol{x}}_0^{(t)} \leftarrow \boldsymbol{x}_0$
11:    $\hat{y}_0^{(t)} \sim p_\theta^{(y)}(\boldsymbol{y}_0 \mid y_t, \hat{\boldsymbol{x}}_0^{(t)}, \boldsymbol{x}_{cor})$
12:    $y_{t-1} \sim q(y_{t-1} \mid y_t, \hat{y}_0^{(t)})$
13: **end for**
14: **return** $y_0$

---

## C.3. Probabilistic Derivation

We derive the probabilistic objective underlying the *Nested* strategy and show how the sampling rule in Algo. 6 follows from a reverse-time factorization combined with point-estimate (Dirac) approximations, in the spirit of standard DDPM derivations (Ho et al., 2020) and nested inner-loop sampling (Elata et al., 2024).

**Conditional reverse-time objective** In the Nested strategy, we treat the logit diffusion as the *outer* generative process. At inference time we aim to sample a logit trajectory from the conditional reverse-time distribution

$$p(y_{0:T-1} \mid y_T, \boldsymbol{x}_{cor}),$$

with $y_T \sim \mathcal{N}(0, I)$. Using the Markov structure of the reverse diffusion, we factorize

$$p(y_{0:T-1} \mid y_T, \boldsymbol{x}_{cor}) = \prod_{t=1}^{T} p(y_{t-1} \mid y_t, \boldsymbol{x}_{cor}). \quad (26)$$

It therefore suffices to derive a tractable approximation to a single transition $p(y_{t-1} \mid y_t, \boldsymbol{x}_{cor})$.

**Single-step marginalization through latent clean signal and logits** To relate the intractable transition to estimable quantities, we introduce the clean signal $\boldsymbol{x}_0$ and clean logits $y_0$ via marginalization:

$$p(y_{t-1} \mid y_t, \boldsymbol{x}_{cor}) = \\ \iint p(y_{t-1} \mid y_t, y_0, \boldsymbol{x}_0, \boldsymbol{x}_{cor}) \cdot \\ p(y_0, \boldsymbol{x}_0 \mid y_t, \boldsymbol{x}_{cor}) \, d\boldsymbol{x}_0 \, dy_0. \quad (27)$$

Applying the chain rule to the joint posterior yields

$$p(y_0, \boldsymbol{x}_0 \mid y_t, \boldsymbol{x}_{cor}) = p(y_0 \mid y_t, \boldsymbol{x}_0, \boldsymbol{x}_{cor}) \, p(\boldsymbol{x}_0 \mid y_t, \boldsymbol{x}_{cor}), \quad (28)$$

and substituting into (27) gives

$$p(y_{t-1} \mid y_t, \boldsymbol{x}_{cor}) = \\ \iint p(y_{t-1} \mid y_t, y_0, \boldsymbol{x}_0, \boldsymbol{x}_{cor}) \cdot \\ p(y_0 \mid y_t, \boldsymbol{x}_0, \boldsymbol{x}_{cor}) \cdot \\ p(\boldsymbol{x}_0 \mid y_t, \boldsymbol{x}_{cor}) \, d\boldsymbol{x}_0 \, dy_0. \quad (29)$$

The main difficulty here lies in the unknown posteriors $p(\boldsymbol{x}_0 \mid y_t, \boldsymbol{x}_{cor})$ and $p(y_0 \mid y_t, \boldsymbol{x}_0, \boldsymbol{x}_{cor})$, which we approximate using point estimates.

**Inner-loop approximation of the clean-signal posterior** The key design choice in the Nested strategy is to approximate the clean-signal posterior $p(\boldsymbol{x}_0 \mid y_t, \boldsymbol{x}_{cor})$ using a *full*

*inner diffusion trajectory* for $x$ conditioned on the current outer state $y_t$. Concretely, define the inner reverse-time process

$$p(x_{0:T-1} \mid x_T, y_t, x_{cor}) =$$
$$\prod_{\tau=1}^{T} p(x_{\tau-1} \mid x_\tau, y_t, x_{cor}), \quad (30)$$

with $x_T \sim \mathcal{N}(0, I)$. In practice, although the target conditional distribution is defined with respect to $(y_t, x_{cor})$, the neural network parameterizing $p_\theta^{(x)}$ additionally receives the current outer estimate $\hat{y}_0^{(t+1)}$ as an auxiliary conditioning input, which serves as semantic guidance for the inner diffusion process. Sampling (30) produces a full inner reverse trajectory $\{x_\tau\}_{\tau=0}^{T}$, whose terminal state serves as an approximation of the clean signal. We therefore define $\hat{x}_0^{(t)} = x_0$, where $x_{\tau-1} \sim p_\theta^{(x)}(x_{\tau-1} \mid x_\tau, \hat{y}_0^{(t+1)}, y_t, x_{\text{cor}})$ for $\tau \in [1, T]$, emphasizing that $\hat{x}_0^{(t)}$ is obtained as the endpoint of a full inner diffusion trajectory rather than a single-step network prediction.

Following the standard point-estimate approximation used in diffusion sampling (Ho et al., 2020), we approximate the posterior over $x_0$ by a Dirac mass centered at this inner-loop output:

$$p(x_0 \mid y_t, \hat{y}_0^{(t+1)} x_{cor}) \approx \delta\left(x_0 - \hat{x}_0^{(t)}\right). \quad (31)$$

Substituting (31) into (29) collapses the integral over $x_0$:

$$p(y_{t-1} \mid y_t, x_{cor}) \approx \int p(y_{t-1} \mid y_t, y_0, \hat{x}_0^{(t)}, x_{cor}) \cdot$$
$$p(y_0 \mid y_t, \hat{x}_0^{(t)}, x_{cor}) \, dy_0. \quad (32)$$

**Point estimate for the clean logits**   We now approximate the remaining posterior of the logits, $p(y_0 \mid y_t, \hat{x}_0^{(t)}, x_{cor})$ by a point mass at a predicted clean-logit estimate $\hat{y}_0^{(t)}$. In our formulation, this estimate is produced by a conditional logit denoiser that takes the current noisy logits $y_t$ and semantic features extracted from both the reconstructed signal and the corrupted observation:

$$\hat{y}_0^{(t)} \sim p_\theta^{(y)}\left(y_0 \mid y_t, \hat{x}_0^{(t)}, x_{cor}\right), \quad (33)$$

and we apply the Dirac approximation

$$p(y_0 \mid y_t, \hat{x}_0^{(t)}, x_{cor}) \approx \delta\left(y_0 - \hat{y}_0^{(t)}\right). \quad (34)$$

Substituting (34) into (32) collapses the remaining integral:

$$p(y_{t-1} \mid y_t, x_{cor}) \approx q\left(y_{t-1} \mid y_t, \hat{y}_0^{(t)}, x_{cor}\right). \quad (35)$$

Importantly, conditioned on $(y_t, \hat{y}_0^{(t)})$, the posterior is independent of $x_{cor}$, which influences the transition only through the estimate $\hat{y}_0^{(t)}$ and is therefore redundant in the conditioning. Formally,

$$q(y_{t-1} \mid y_t, \hat{y}_0^{(t)}, x_{cor}) = q(y_{t-1} \mid y_t, \hat{y}_0^{(t)}),$$

see Appendix A for details.

**Connection to Algo. 6**   Equation (35) is exactly the outer-loop sampling rule used in lines 11-12 of Algo. 6: at each reverse-time step $t$ of the logit diffusion, we (i) run an inner $T$-step signal diffusion trajectory conditioned on $y_t$, $\hat{y}_0^{(t+1)}$ and $x_{cor}$ to obtain $\hat{x}_0^{(t)}$ (lines 4-10), (ii) predict $\hat{y}_0^{(t)}$ using (33) (line 11), and (iii) sample $y_{t-1}$ from the Gaussian posterior (35) (line 12). Repeating this procedure for $t = T, \ldots, 1$ and combining with the factorization in (26) yields the trajectory-level objective stated in Proposition 3.2.

### C.4. Training Procedure

Training of the Nested strategy follows the same joint noise-prediction objective as the other coupling schemes, while explicitly reflecting the label-centric structure of the inference procedure. In particular, signal diffusion is treated as an auxiliary process whose role is to provide high-fidelity conditioning for logit denoising.

Given a clean sample $x_0$ and its corrupted observation $x_{cor}$, we first obtain clean logits $y_0 = f_\phi(x_0)$. To emulate the nested inference dynamics, we generate a clean signal estimate $\hat{x}_0$ by running a full reverse diffusion trajectory conditioned on $x_{cor}$. This estimate is treated as a fixed conditioning variable during optimization.

At each training step, a diffusion timestep $t$ is sampled uniformly and Gaussian noise is injected to produce $(x_t, y_t)$. The logit denoiser is trained to predict the injected noise conditioned on the signal estimate $\hat{x}_0$. The resulting logit prediction is then converted into a clean logit estimate $\hat{y}_0$, which is subsequently used to condition the signal denoiser. The training loss is defined as the sum of the signal and logit noise-prediction errors.

This procedure aligns the training distribution with the nested inference structure, ensuring that logit transitions are consistently guided by high-fidelity signal reconstructions. The complete training algorithm is summarized in Algo. 7.

## D. Image Experiment Details

### D.1. Classifier Architectures

For all image experiments, we employ dataset-specific convolutional classifiers trained *exclusively on clean images*.

**Algorithm 7** Nested Training Algorithm

---

1: **repeat** until convergence
2:   Sample $(\boldsymbol{x}_0, \boldsymbol{x}_{cor})$ from $\mathcal{D}$
3:   $\boldsymbol{y}_0 \leftarrow f_\phi(\boldsymbol{x}_0)$
4:   $\hat{\boldsymbol{x}}_0 \leftarrow$ Full diffusion sampling conditioned on $\boldsymbol{x}_{cor}$
5:   *// Training iterations*
6:   **for** $k = 1$ **to** $K$ **do**
7:     Sample $t \sim \text{Uniform}(\{1, \ldots, T\})$
8:     Sample $\boldsymbol{\epsilon}_x, \boldsymbol{\epsilon}_y \sim \mathcal{N}(0, \boldsymbol{I})$
9:     $\boldsymbol{x}_t \sim q(\boldsymbol{x}_t \mid \boldsymbol{x}_0, \boldsymbol{\epsilon}_x), \boldsymbol{y}_t \sim q(\boldsymbol{y}_t \mid \boldsymbol{y}_0, \boldsymbol{\epsilon}_y)$
10:     $\hat{\boldsymbol{\epsilon}}_y \leftarrow \text{Denoiser}_\theta^{(y)}(\boldsymbol{y}_t, \hat{\boldsymbol{x}}_0, \boldsymbol{x}_{cor}, t)$
11:     $\hat{\boldsymbol{y}}_0 \leftarrow q(\boldsymbol{y}_0 \mid \boldsymbol{y}_t, \hat{\boldsymbol{\epsilon}}_y)$ *// Estimate from diffusion equation*
12:     $\hat{\boldsymbol{\epsilon}}_x \leftarrow \text{Denoiser}_\theta^{(x)}(\boldsymbol{x}_t, \hat{\boldsymbol{y}}_0, \boldsymbol{x}_{cor}, t)$
13:     $\mathcal{L} \leftarrow \|\hat{\boldsymbol{\epsilon}}_x - \boldsymbol{\epsilon}_x\|_2^2 + \|\hat{\boldsymbol{\epsilon}}_y - \boldsymbol{\epsilon}_y\|_2^2$
14:     Apply optimization step on $\mathcal{L}$ to update $\theta$
15:   **end for**
16: **end repeat**

---

These classifiers are used to produce semantic logits $\boldsymbol{y}$ and are kept *frozen* throughout diffusion-based training and inference.

For CIFAR-10, CIFAR-100, and ImageNet32-100, we adopt ResNet-50 backbones adapted for $32 \times 32$ inputs by replacing the initial $7 \times 7$ convolution with a $3 \times 3$ kernel and removing the initial max-pooling layer. This modification preserves spatial resolution and is standard for small-image benchmarks.

For MNIST, we use a lightweight convolutional classifier operating on grayscale $28 \times 28$ images, consisting of three convolutional layers with ReLU activations and interleaved max-pooling, followed by two fully connected layers producing 10-class logits. This architecture is sufficient to achieve strong performance on clean MNIST and is kept fixed throughout all experiments.

All classifiers are trained using standard cross-entropy loss on clean training data and evaluated on clean validation sets to ensure strong baseline performance. During all diffusion experiments, classifier parameters remain fixed and are never updated. This isolates the effect of the proposed framework, ensuring that performance improvements arise from the coupled diffusion dynamics over the signal and logits, rather than from any adaptation of the classifier itself.

### D.2. Diffusion Model Architecture

Our framework comprises two learnable denoisers: (i) a signal denoiser operating in the input space, and (ii) a logits denoiser operating in the semantic space. Both denoisers are time-conditioned and are trained to predict the additive

Gaussian noise in their respective diffusion processes, from which the reverse-time updates are computed analytically during sampling. Throughout all experiments, the downstream classifier remains frozen; semantic information enters the diffusion model only through the evolving logits variables.

**Signal denoiser.** For all image datasets, we use a compact U-Net architecture with residual bottleneck blocks and multi-scale encoder-decoder skip connections. Timestep information is injected via a learned embedding followed by an MLP that produces per-channel biases added to intermediate feature maps. To support conditional generation, the U-Net optionally includes additional encoder streams for the corrupted observation and for semantic conditioning (logits), whose multi-scale features are fused into the main decoder via concatenation at the bottleneck and skip levels. This design enables the signal denoiser to leverage semantic context while retaining fine spatial structure through skip connections.

**Logits denoiser.** The logits denoiser predicts the reverse diffusion update for $\boldsymbol{y}_t$ conditioned on the current noisy logits and on image features extracted from the current signal state. For CIFAR-10 and MNIST, we employ a lightweight conditional MLP denoiser, where an auxiliary CNN encoder (LeNet-style) extracts a low-dimensional representation from the image, and timestep conditioning is implemented via FiLM-like feature modulation. For the larger class spaces (CIFAR-100 and ImageNet32-100), we use a higher-capacity conditional architecture: a ResNet-18 image encoder produces a global feature vector, time is encoded with sinusoidal embeddings, and the logits denoising network combines FiLM-style conditioning with cross-attention between logit features and image features. This allows the semantic diffusion process to adapt its denoising trajectory based on the evolving signal content while scaling to higher-dimensional logits.

### D.3. Training Configuration

All diffusion models are trained using a cosine noise schedule. For both the signal and logits diffusion processes, we employ a standard mean squared error (MSE) objective corresponding to the prediction of the reverse-time update at each diffusion step.

To improve training stability, we adopt a staged training procedure. Specifically, the signal denoiser and the logits denoiser are first trained independently for a small number of epochs using their respective single-modality formulations, conditioned only on their own noisy variables. During this initialization phase, cross-modal conditioning is disabled. The fully coupled training objective is introduced only after both denoisers have converged to a rea-

sonable solution. This warm-start strategy provides a stable initialization and empirically prevents early training instabilities caused by unreliable cross-modal guidance at high noise levels.

### D.4. Sampling Procedure

All diffusion models are sampled using a deterministic DDIM sampler with a fixed budget of 150 reverse-time steps. Unless stated otherwise, the same sampling configuration is used across datasets and methods to ensure a fair comparison.

**Parallel strategy.** In the *Parallel* strategy, signal and logits are initially denoised independently. Specifically, during the first half of the reverse diffusion trajectory, each modality is sampled without conditioning on the other. Conditioning is enabled only in the later stages of sampling, once both the signal and logits have reached a sufficiently informative regime. This warm-up phase mitigates instability caused by early-time noise, where mutual guidance is unreliable, and empirically leads to more stable and accurate sampling.

**Alternating strategy.** In the *Alternating* strategy, sampling proceeds in multiple macro-iterations. Each iteration consists of a full reverse diffusion of the signal followed by a full reverse diffusion of the logits. The prediction produced by one modality in the current iteration is used as conditioning input for the other modality in the subsequent iteration. In our experiments, we use five such alternating iterations, allowing information to be progressively exchanged between modalities while maintaining fully coupled trajectories.

**Nested strategy.** The *Nested* strategy employs a hierarchical sampling schedule in which the signal serves as a guiding variable for logits diffusion. We first perform a full reverse diffusion of the signal to obtain an initial estimate of $x_0$, which is then used to condition the logits diffusion process. During the early stages of logits diffusion, this signal estimate is held fixed and the signal is not updated. After an initial warm-up of 50 steps, the signal estimate is periodically refreshed by running a full signal diffusion every 20 sampling steps, and the updated $x_0$ is subsequently used to guide the remaining logits diffusion. This design balances computational efficiency with the ability to iteratively refine the signal guidance as semantic estimates evolve.

**Computational cost (NFE).** We compare sampling efficiency across strategies in terms of the number of neural function evaluations (NFE), abstracting away hardware-specific runtime effects. We distinguish between NFE in-

curred by the diffusion models (denoisers) and NFE incurred by the downstream classifier used for semantic guidance.

In the *Parallel* strategy, signal and logits are each sampled once via a full diffusion trajectory, resulting in $2\times$ the NFE of a standard single-modality diffusion model. In addition, the classifier is evaluated once for initialization and subsequently at every reverse step during the coupled phase, in which the signal and logits are mutually conditioned on each other(second half of the trajectory), yielding approximately $1 + T/2$ classifier evaluations.

The *Alternating* strategy performs a full signal diffusion followed by a full logits diffusion at each macro-iteration; with five alternating iterations, this amounts to $10\times$ the baseline denoiser NFE. The classifier is evaluated once for initialization and once after each completed signal diffusion, resulting in a total of $1 + 5$ classifier calls.

Finally, the *Nested* strategy requires one initial signal diffusion to initialize $x_0$, followed by five additional signal refinements during logits sampling, together with a single logits diffusion, yielding a total of $7\times$ the baseline denoiser NFE. Similarly to the Alternating strategy, the classifier is evaluated once for initialization and after each signal update $\hat{x}_0^{(t)}$, for a total of $1 + 5$ classifier calls.

Finally, we note that NFE for the signal, logits, and classifier are not directly comparable in practice, as they operate in different spaces and may incur different computational costs per evaluation.

### D.5. Qualitative Signal Enhancement Examples

To complement the quantitative results, we provide qualitative examples illustrating how the proposed coupled strategies enhance corrupted signals in a way that facilitates correct classification. Fig. 5 shows MNIST samples under corruption, together with the outputs of the Enhanced baseline and our three coupled strategies.

While the Enhanced baseline often produces visually plausible reconstructions, it tends to preserve ambiguous or distorted structures that remain difficult to classify. In contrast, the *Alternating*, *Parallel*, and *Nested* strategies consistently recover more coherent digit shapes, removing spurious artifacts and reinforcing class-discriminative features. This qualitative behavior aligns with the observed accuracy improvements and highlights the role of semantic guidance in steering the signal denoising process toward classification-relevant solutions rather than purely perceptual ones.

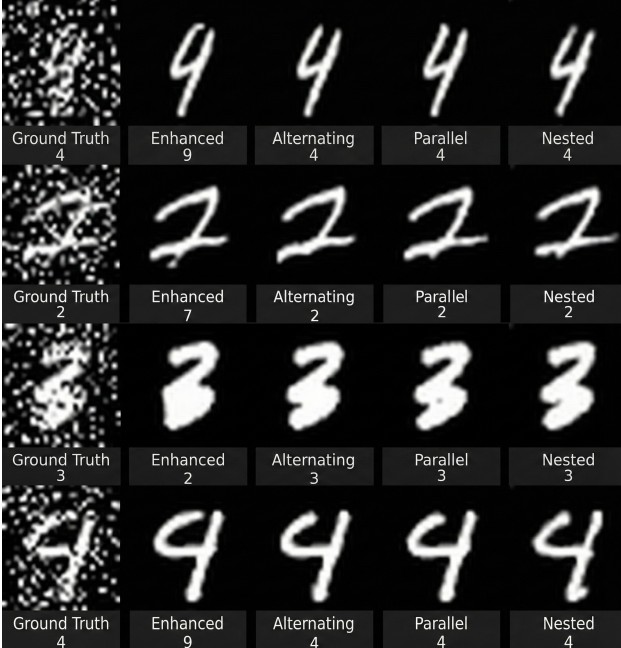

*Figure 5.* Qualitative signal enhancement examples on MNIST. Each row corresponds to a different test sample. From left to right: corrupted input (ground truth shown for reference), Enhanced baseline, Alternating, Parallel, and Nested strategies. The predicted class under each reconstruction is indicated in the figure. Our coupled strategies consistently recover digit structures that are visually clearer and semantically aligned with the true class, leading to correct predictions.

## E. Comparison of Inference Strategies

Each strategy reflects a different trade-off between compute, stability, and accuracy, but more fundamentally, they differ in how tightly and adaptively the signal and semantic processes interact. *Parallel* performs step-level coupling, exchanging intermediate estimates at every timestep. This enables continuous bidirectional feedback, allowing errors to be corrected early and progressively refined, which leads to fast convergence and strong performance on complex datasets. Its use of evolving estimates also implicitly captures uncertainty, making the interaction more adaptive as the diffusion progresses. In practice, it is also the fastest strategy. *Alternating* operates at a coarser granularity, conditioning each diffusion block on a single fixed endpoint estimate. While this makes it simple and stable, it commits to a single hypothesis without exposing uncertainty, limiting its ability to recover from early mistakes. As a result, it scales poorly to more complex settings, where fixed conditioning becomes a bottleneck and slows convergence. It is also the slowest strategy in practice. *Nested* strengthens the coupling by recomputing the signal trajectory for each logit update, enabling more consistent alignment between the two processes and improved robustness in challenging regimes. Empirically, it achieves the strongest overall performance on the most complex datasets and in terms of runtime, it lies between the two.

In practice, Parallel serves as a strong default due to its efficient and adaptive interaction, Alternating is suitable for simpler or more stable regimes, and Nested is preferable when maximizing robustness and accuracy is the primary objective.

## F. Audio Experiment Details

### F.1. Signal Diffusion for Audio: SGMSE in the Complex STFT Domain

**Overview and motivation.** For speech enhancement experiments, we instantiate the diffusion component using *Score-Based Generative Models for Speech Enhancement (SGMSE)* (Richter et al., 2023). Unlike the discrete-time DDPM formulation used in our image experiments, SGMSE operates in continuous time and is defined directly in the *complex STFT domain*. This formulation has been shown to be particularly effective for speech denoising and dereverberation, as it incorporates the noisy observation directly into the diffusion process via a task-adapted stochastic differential equation (SDE).

**STFT representation.** Given a clean waveform $s \in \mathbb{R}^L$, we compute its complex short-time Fourier transform. Following SGMSE, we operate directly in the complex STFT domain by representing each time-frequency bin via its real and imaginary parts, which together form the signal state $x_0$. The corrupted observation $x_{cor}$ is obtained by applying the same STFT representation to the noisy or reverberant input signal and is used to condition the diffusion process. After enhancement, the estimated clean representation is mapped back to the time domain using the inverse STFT.

**Task-adapted forward SDE.** SGMSE defines a forward noising process that explicitly models the corruption process rather than diffusing clean speech toward pure Gaussian noise. Let $x_t$ denote the latent clean-speech STFT representation at continuous diffusion time $t \in [0, 1]$. The forward process is defined as

$$d\boldsymbol{x}_t = \gamma\big(\boldsymbol{x}_{cor} - \boldsymbol{x}_t\big)\, dt + g(t)\, d\boldsymbol{w}_t, \qquad (36)$$

where $\boldsymbol{w}_t$ is standard Brownian motion, $\gamma > 0$ is a stiffness parameter controlling the attraction toward the corrupted observation, and $g(t)$ follows a variance-exploding (VE) noise schedule. This drift term ensures that the mean of the forward process transitions smoothly from clean speech toward the corrupted signal $\boldsymbol{x}_{cor}$, which distinguishes SGMSE from unconditional score-based diffusion models (Song et al., 2020b).

**Conditional reverse-time SDE.** Speech enhancement is performed by simulating the reverse-time SDE associated with the forward process:

$$d\boldsymbol{x}_t = \Big[\gamma(\boldsymbol{x}_{cor} - \boldsymbol{x}_t)$$
$$- g(t)^2 \nabla_{\boldsymbol{x}_t} \log p_t(\boldsymbol{x}_t \mid \boldsymbol{x}_{cor})\Big] dt + g(t)\, d\bar{\boldsymbol{w}}_t. \tag{37}$$

where $\bar{\boldsymbol{w}}_t$ denotes Brownian motion in reverse time. The conditional score $\nabla_{\boldsymbol{x}_t} \log p_t(\boldsymbol{x}_t \mid \boldsymbol{x}_{cor})$ is approximated by a neural network $s_\theta(\boldsymbol{x}_t, t, \boldsymbol{x}_{cor})$, which is trained to estimate the score of the perturbation kernel induced by the forward SDE.

**Training via denoising score matching.** Since the forward SDE admits a closed-form Gaussian perturbation kernel, states $\boldsymbol{x}_t$ can be sampled directly. We first define the mean trajectory induced by the task-adapted drift as

$$\boldsymbol{\mu}(\boldsymbol{x}_0, \boldsymbol{x}_{cor}, t) = e^{-\gamma t}\boldsymbol{x}_0 + \big(1 - e^{-\gamma t}\big)\boldsymbol{x}_{cor}. \tag{38}$$

The latent state is then obtained by sampling $\boldsymbol{x}_t = \boldsymbol{\mu}(\boldsymbol{x}_0, \boldsymbol{x}_{cor}, t) + \sigma(t)\boldsymbol{z}$, where $\boldsymbol{z} \sim \mathcal{N}(0, \boldsymbol{I})$.

SGMSE trains the score network using denoising score matching. This involves regressing the model output $s_\theta(\boldsymbol{x}_t, t, \boldsymbol{x}_{cor})$ toward the analytically known score target $-\boldsymbol{z}/\sigma(t)$. Importantly, the network is trained to predict the *score* rather than additive noise, in contrast to the DDPM-style parameterization used in our image experiments.

**Sampling.** At inference time, clean speech is generated by solving the reverse-time SDE starting from an initial condition $\boldsymbol{x}_T \sim \mathcal{N}(\boldsymbol{x}_{cor}, \sigma(T)^2\boldsymbol{I})$, i.e., a heavily perturbed version of the corrupted observation. Following SGMSE, we employ a predictor-corrector sampler consisting of a discretized reverse-SDE step followed by a small number of Langevin correction steps. All sampler hyperparameters are reported in Appendix F.5.

**Relation to our coupled framework.** In our coupled diffusion framework, SGMSE defines the *signal diffusion backbone* for audio. Additional semantic variables, such as ASR logits, are incorporated solely as auxiliary conditioning inputs to the score network $s_\theta$, depending on the chosen coupling strategy. Crucially, these additions do not alter the underlying SGMSE formulation: the forward SDE, reverse-time dynamics, noise schedule, and sampling procedure remain unchanged. Thus, all coupling strategies share the same SGMSE-based signal diffusion process and differ only in how semantic information is injected into the score model.

**Logit diffusion counterpart.** The same diffusion formulation is applied to the classifier logits. Given a clean signal $\boldsymbol{x}_0$, we define the clean semantic target as $\boldsymbol{y}_0 = f_\phi(\boldsymbol{x}_0)$, where $f_\phi$ denotes the frozen downstream classifier, and similarly $\boldsymbol{y}_{cor} = f_\phi(\boldsymbol{x}_{cor})$ for the corrupted input. Diffusion is then performed directly in the *logit space* $\boldsymbol{y}$, rather than in the STFT domain. The forward and reverse processes, training objectives, and coupling mechanisms are analogous to those used for the signal diffusion, with the key distinction that the state variables correspond to logits and evolve in a semantic representation space instead of a time-frequency representation.

### F.2. Datasets and Preprocessing

**Google Speech Commands** The Google Speech Commands (GSC) dataset is a standard benchmark for keyword spotting, consisting of one-second utterances of isolated words recorded at 16 kHz by a large and diverse set of speakers. We use a subset of ten commands: *down, go, left, no, off, on, right, stop, up,* and *yes,* forming a closed-set speech classification task.

**Reverberation Augmentation.** To simulate realistic acoustic conditions, we generate a reverberant version of the dataset using physics-based room simulations. Each clean utterance is convolved with a randomly sampled room impulse response (RIR) generated using `pyroomacoustics`. Rooms are modeled as shoebox environments with randomly sampled dimensions and reverberation times (RT60 uniformly drawn from 0.25-0.8 s). Source and microphone positions are randomly placed within the room, subject to minimum distance constraints.

**EARS** The EARS (Expressive Anechoic Recordings of Speech) dataset is a large-scale, high-quality speech corpus recorded in an anechoic chamber at 48 kHz, comprising approximately 100 hours of speech from over 100 speakers with high demographic and expressive diversity (Richter et al., 2024). The dataset includes multiple speaking styles and content types, ranging from phonetically balanced read sentences to emotional speech, conversational free-form speech, and non-verbal vocalizations.

A substantial portion of EARS consists of *known sentences* drawn from predefined scripts, alongside *free-form speech* where speakers produce unconstrained content. In this work, we train our models exclusively on the known-sentence subset. This restriction reduces the effective vocabulary of the downstream ASR system to 838 tokens, simplifying the diffusion process over logits. Extending the method to unrestricted vocabularies would require larger training corpora, as in large-scale ASR systems, or alternatively performing diffusion in a lower-dimensional embedding space of the logits.

**EARS-Reverb.** For dereverberation experiments, we construct EARS-Reverb following the procedure of the original EARS benchmark (Richter et al., 2024). Clean anechoic speech is convolved with real room impulse responses (RIRs) collected from multiple public datasets, spanning a wide range of acoustic environments. The RIRs are aligned to remove direct-path delays and restricted to moderate reverberation times, producing realistic reverberant speech while preserving temporal alignment with the clean signal.

**EARS-WHAM.** For speech enhancement under additive noise, we use EARS-WHAM, where clean EARS utterances are mixed with real noise recordings from the WHAM! dataset, following the protocol described in (Richter et al., 2024). Speech and noise are mixed at randomly sampled signal-to-noise ratios, producing realistic noisy speech conditions.

**Segmentation.** All EARS-based audio is segmented into approximately 4-second chunks. Segmentation is performed using a voice activity detection (VAD) model to avoid cuts in the middle of words or phonetic units, ensuring linguistically coherent segments suitable for ASR-based evaluation.

### F.3. Whisper Guidance and Evaluation

We use Whisper as the downstream ASR model for both guidance and evaluation. Whisper is a sequence-to-sequence transformer-based ASR system trained on large-scale multilingual speech data. In all experiments, we use the *Whisper-base* model, which provides a favorable trade-off between recognition accuracy and computational cost.

During decoding, Whisper generates the transcription autoregressively, predicting one token at a time. At each decoding step, the model produces a logit vector over the full vocabulary. For our framework, we extract and retain only the logits corresponding to a predefined subset of tokens relevant to the task, and use these vectors as the logits input to our diffusion model. Importantly, while Whisper itself computes representations over the full vocabulary, the final transcription is formed by predicting tokens strictly from the predefined subset.

Crucially, this restricted-token decoding is applied consistently across all methods and baselines. This ensures that all approaches operate under the same output space and evaluation protocol, enabling a fair and controlled comparison that isolates the effect of the different enhancement and coupling strategies.

For evaluation, we compute the word error rate (WER) using task-specific references. On the Google Speech Commands dataset, WER is computed by comparing the pre-

dicted transcription to the ground-truth command label. On the EARS dataset, WER is computed by comparing the transcription obtained from enhanced or dereverberated speech to the transcription produced by Whisper on the corresponding clean audio, following standard practice (Richter et al., 2024).

### F.4. Diffusion Models Architecture

**Signal denoiser.** For the audio signal diffusion component, we adopt the SGMSE architecture and operate in the complex STFT domain, using an NCSN++ U-Net backbone as in the original SGMSE implementation. Specifically, for Google Speech Commands we use the standard `NCSNpp` configuration, whereas for EARS we use the 48 kHz variant `NCSNpp_48k` to match the sampling rate and model configuration used in the EARS benchmarks.

In our coupled setup, the signal denoiser receives the same inputs as in SGMSE (noised complex STFT and noise-level embedding), and we additionally provide *logits-derived conditioning* to guide the denoising process. Concretely, we inject this conditioning into the U-Net bottleneck via a lightweight projection and modulation mechanism (FiLM-style in Google Speech Commands, and a mid-level spatial fusion in EARS), without modifying the underlying SGMSE forward noising process or the SDE/score-matching formulation.

**Initialization from pretrained SGMSE models.** We do not train the signal denoiser from scratch. Instead, we initialize from pretrained checkpoints released with the SGMSE implementation, which reduces optimization difficulty (analogous to our image experiments). For Google Speech Commands, we initialize from an `NCSNpp` model pretrained on WSJ0-REVERB. For EARS-Reverb, we initialize from an `NCSNpp_48k` model pretrained on EARS-Reverb, and for EARS-WHAM we initialize from the corresponding `NCSNpp_48k` checkpoint pretrained on EARS-WHAM. We then fine-tune these pretrained signal denoisers under our coupled training objective.

**Logits denoiser.** We parameterize the reverse-time update in logit space using a dedicated *logits denoiser* that operates on the current noised logits and auxiliary guidance signals.

The denoiser predicts *a single logit vector at each decoding step*, rather than the entire sequence jointly, and is conditioned on a limited history of previously generated logits from the decoding sequence. In addition, the logits denoiser is explicitly conditioned on *audio-derived features* obtained by applying the Whisper encoder to the input audio. These features are further processed using a lightweight Transformer and attention-based pooling to

provide acoustically aligned guidance for each decoding step, incorporating both linguistic context and acoustic information. For Google Speech Commands, we employ a compact MLP-based score network with sinusoidal time embeddings, FiLM-conditioned residual blocks, and optional attention to capture dependencies between classes. For EARS, where the logits correspond to a large Whisper vocabulary, we use a higher-capacity architecture tailored to autoregressive decoding.

In both settings, we first train the logits denoiser independently on logits alone to improve stability and ensure fair initialization, and only then perform coupled training with the signal diffusion model.

### F.5. Training and Sampling Hyperparameters

**Signal diffusion.** All audio experiments are trained using the stochastic differential equation (SDE) formulation adopted in SGMSE. We use the standard SGMSE training objective, where the neural network predicts the score function scaled by the noise level, and the loss is defined with respect to the injected Gaussian noise. All SDE parameters (noise schedule, minimum and maximum noise levels, and discretization settings) are taken directly from the original SGMSE implementation.

Prior to training, audio waveforms are normalized to the range $[-1, 1]$. To ensure consistency between signal and logit diffusion, logits are normalized to have the same mean and standard deviation as the audio inputs, as the SDE hyperparameters are calibrated for audio diffusion.

At inference time, we discretize the reverse-time SDE and use 50 sampling steps for all audio experiments. Each step consists of one predictor step followed by one corrector step. For the predictor, we use the `ReverseDiffusionPredictor` as in the default SGMSE setup. For the corrector, we apply `AnnealedLangevinDynamics` with a target signal-to-noise ratio of 0.5 to determine the adaptive step size.

**Logit diffusion.** For diffusion in logit space, the neural network does not predict the full score directly. Instead, it predicts a *residual correction* relative to an analytic baseline score derived from the noisy logits. The true score is defined via the mean trajectory $\boldsymbol{\mu}(\boldsymbol{y}_0, \boldsymbol{y}_{\text{cor}}, t) = e^{-\gamma t}\boldsymbol{y}_0 + \left(1 - e^{-\gamma t}\right)\boldsymbol{y}_{\text{cor}}$ as

$$\hat{s}_t^{(0)}(\boldsymbol{y}_t, \boldsymbol{y}_0, \boldsymbol{y}_{\text{cor}}) = -\frac{\boldsymbol{y}_t - \boldsymbol{\mu}(\boldsymbol{y}_0, \boldsymbol{y}_{\text{cor}}, t)}{\sigma_t^2}. \quad (39)$$

Since the corrupted logits $\boldsymbol{y}_{\text{cor}}$ provide a strong initialization, we use the corresponding baseline

$$\hat{s}_t^{(0)}(\boldsymbol{y}_t, \boldsymbol{y}_{\text{cor}}) = -\frac{\boldsymbol{y}_t - \boldsymbol{y}_{\text{cor}}}{\sigma_t^2}, \quad (40)$$

and train the logit denoiser to predict only the residual between this baseline and the true score. This residual formulation simplifies the learning problem and improves training stability, particularly for high-dimensional logit spaces.

**Sampling Procedure.** Sampling in the audio experiments follows the same strategy definitions used for images, with minor adaptations. In the *Parallel* strategy, signal and logits are sampled independently for most of the reverse-time trajectory, and coupling between modalities is enabled only during the final 10 steps, once both have reached a lower-noise regime. In the *Alternating* strategy, we alternate between full signal diffusion and full logit diffusion passes, using the output of one modality to condition the other in the next iteration; in all audio experiments, we use 5 such alternating iterations. In the *Nested* strategy, we first perform a full reverse diffusion of the audio signal to obtain an initial estimate of $\boldsymbol{x}_0$, which is then used to condition the logits diffusion. During the final 5 steps, the signal diffusion is re-run to update $\boldsymbol{x}_0$, allowing the signal estimate to be refined as the semantic content of the logits becomes more reliable.

**Computational cost (NFE).** As discussed for the image experiments, the strategies differ in sampling cost due to the number of full diffusion trajectories they require. The *Parallel* strategy performs one full signal diffusion and one full logit diffusion, resulting in $2\times$ the baseline denoiser NFE. In addition, the classifier is evaluated once for initialization and subsequently at every reverse step during the coupled phase, in which the signal and logits are mutually conditioned (final 10 steps), yielding $1 + 10$ classifier evaluations.

The *Alternating* strategy runs a full signal and logit diffusion at each iteration; with five iterations, this incurs $10\times$ the baseline denoiser NFE. The classifier is evaluated once for initialization and once after each signal diffusion, resulting in a total of $1 + 5$ classifier calls.

The *Nested* strategy performs one initial signal diffusion, followed by five additional signal refinements during logits sampling and a single logits diffusion, yielding $7\times$ the baseline denoiser NFE. Similarly, the classifier is evaluated once for initialization and after each signal update, for a total of $1 + 5$ classifier calls.

