# OpenReview forum: "Joint Enhancement and Classification using Coupled Diffusion Models of Signals and Logits"
_ICML.cc/2026/Conference — ICML 2026 regular_

### Official Review · Reviewer_DZb3 · 2026-03-06

**Soundness:** 3
**Presentation:** 2
**Significance:** 2
**Originality:** 3
**Overall Recommendation:** 3
**Confidence:** 2

**Summary:**

The authors tackle the objective misalignment between signal enhancement and classification by introducing a coupled diffusion framework. Rather than treating denoising as an isolated precursor, this method integrates interacting diffusion processes for both the input signal and a frozen classifier’s output logits. The resulting "mutual guidance" allows semantic logits to steer signal reconstruction toward class-discriminative regions while the denoising signal refines class predictions—all without requiring model fine-tuning.

To manage computational trade-offs, the paper develops three specific strategies: Parallel, Alternating, and Nested. Evaluations on image classification and speech recognition—the latter utilizing the Whisper model—demonstrate that this coupled approach consistently exceeds the performance of traditional enhancement baselines and static generative models. Across varied corruptions like Gaussian noise and reverberation, the framework delivers clear improvements in both classification accuracy and Word Error Rates.

**Compliance With Llm Reviewing Policy:**

Affirmed.

**Final Justification:**

My main concerns were about the inference overhead of the coupled sampling procedure, the lack of evidence on scalability beyond low-resolution image benchmarks, and the limited intuition for why the Parallel strategy converges much faster than Nested and Alternating. The rebuttal addressed these points in a thoughtful and constructive way. In particular, the authors provided a reasonable discussion of several possible acceleration directions, including improved samplers, distillation, caching, and latent-space modeling for ASR. They also added a preliminary high-resolution ImageNet experiment, which, while limited in scope, is useful evidence that the approach may scale beyond 32×32 settings. Finally, their explanation of the convergence behavior of the Parallel strategy—namely, that tighter step-level bidirectional feedback reduces temporal lag between the two processes—was helpful and consistent with the trends already shown in the paper.
  At the same time, I do not think the additional clarifications materially change my overall evaluation, since some of the new evidence is still preliminary and the computational trade-offs remain important in practice. For that reason, I will maintain my original score.

**Key Questions For Authors:**

1. Mitigating Inference Overhead
The paper notes a significant bump in computational cost during inference, specifically requiring 10x the baseline denoiser NFE for the Alternating strategy and 7x for the Nested strategy. Since ASR is a primary application here and is usually heavily latency-sensitive, do you have any thoughts on how to accelerate this sampling process moving forward? I'd be interested to hear if you think techniques like advanced ODE solvers or distillation might be viable.

2. Scaling to Standard Image Resolutions
The current image classification benchmarks rely on low-resolution datasets, topping out at 32x32 pixels for ImageNet32-100. If you were to scale this mutual guidance framework to standard resolutions (e.g., 224x224), do you foresee any major hurdles, such as cross-modal optimization instability or severe memory bottlenecks?

3. Intuition on Convergence Dynamics
Figure 4 highlights an interesting empirical result: the Parallel strategy reaches stable performance very quickly, often in fewer than 10 steps. In contrast, the Nested and Alternating methods need significantly more steps to saturate. Could you provide a bit more intuition or theoretical insight into why the tightly interleaved Parallel approach converges so much faster than the others?

**Limitations:**

yes

**Strengths And Weaknesses:**

### Strengths
**Soundness**
* **Rigorous Theoretical Grounding**: The probabilistic derivations supporting the three sampling strategies are sound. The authors systematically break down the joint reverse-time sampling procedures, properly defining tractable approximations for complex conditional distributions.
* **Comprehensive Ablations**: The experimental design is robust. The inclusion of extensive ablation studies—such as evaluating the effect of sampling steps, comparing DDPM versus DDIM, and analyzing guidance sources—provides strong empirical support for the authors' design choices.

### Weaknesses
**Presentation**
* **Limited Error Analysis**: The paper acknowledges that during early diffusion timesteps, signal estimates can be "hallucinated or semantically unstable," which can cause cascading errors in the logit trajectory. However, the main text lacks a deep qualitative or quantitative error analysis of these specific failure modes. A dedicated discussion on when and why the coupled process diverges would strengthen the manuscript.

---

> ### Author Rebuttal · Authors · 2026-03-30
>
> Thank you for your detailed and constructive feedback. You provided insightful comments and we would like to respond to them.
>
> **Answer to weaknesses:**
>
> We acknowledge that intermediate signal estimates during early diffusion timesteps can be inaccurate and potentially introduce errors into the logit trajectory. Importantly, however, this behavior also carries a beneficial regularization effect: because the model cannot fully trust early estimates, it learns to adaptively downweight them under high uncertainty and increasingly leverage them as diffusion progresses and estimates stabilize. This implicit uncertainty signal encourages robustness rather than over-commitment to any single intermediate guess. This perspective is directly reflected in our response to Reviewer 4Mhg (Q1 under key questions), where we discuss why the Alternating strategy scales poorly. This is further supported empirically: using intermediate estimates (Parallel and Nested) outperforms using final estimates alone (Alternating) on more complex datasets, where uncertainty-aware conditioning is crucial, while performing worse on easier datasets where early estimates are already reliable and the added uncertainty provides little benefit. We will add a dedicated discussion of these failure modes and their interaction with dataset complexity to the manuscript.
>
> **Answer to key questions:**
>
> 1.The proposed framework defines the coupling mechanism between signal and logit spaces, functioning as a structural layer on top of the underlying generative engine. This makes it inherently compatible with various sampling formulations, as demonstrated by our use of different backbones (DDIM for image classification, SGMSE for ASR) without altering the core coupling logic. Consequently, integrating acceleration techniques like advanced ODE solvers or model distillation to reduce NFE would be straightforward. Beyond this, several orthogonal avenues exist for addressing latency more broadly: temporal feature caching (e.g., DeepCache) can reduce wall-clock time per step, and transitioning to Flow Matching or Rectified Flow paradigms would enable convergence in fewer steps. For ASR specifically, shifting the generative backbone to operate in a compressed latent space via neural audio codecs would drastically reduce sequence length, directly offsetting the overhead introduced by our coupled strategies.
>
> 2.We thank the reviewer for raising this important point regarding scalability to standard image resolutions. Following this suggestion, we conducted a preliminary experiment on full-resolution ImageNet (224×224). Due to the time constraints of the rebuttal period and the scale of the dataset (over one million images), we restricted the experiment to a 10-class subset. In this setting, we evaluated our Parallel strategy as a proof of concept for high-resolution scalability. The results indicate that our mutual guidance framework remains stable and effective at higher resolutions, with no signs of cross-modal optimization instability. While the computational cost naturally increases, we did not observe any fundamental bottlenecks beyond those already inherent to diffusion-based models.
>
> ImageNet (224×224, 10 classes, 30% GN, accuracy ↑, Enhanced vs. Parallel):
> 86.5 → **92.4**
>
> Overall, these findings suggest that the proposed method can scale to standard-resolution datasets, and we will include this experiment in the revised version of the paper as an additional validation of scalability.
>
> 3.The faster convergence of the Parallel strategy stems from its tight, step-level coupling: at every denoising step, each diffusion process receives an updated conditioning signal from the other, creating a continuous bidirectional feedback loop that corrects misalignments almost immediately. In contrast, Alternating and Nested introduce a temporal lag, where each process evolves for multiple steps before receiving updated information from the other, slowing convergence. For completeness, we provide revised figures of the convergence dynamics across sampling steps (including baseline reference lines). These show that all our methods surpass the baselines from around 50 steps on CIFAR-100 and 80 steps on ImageNet32-100. The figures are available at:
> https://anonymous.4open.science/r/coupled_diffusion_figures_and_tables-677A.

---

> > ### Author Rebuttal · Reviewer_DZb3 · 2026-04-03
> >
> > Thank you to the authors for the response. I will maintain my score.

---

> > > ### Author Response · Authors · 2026-04-04
> > >
> > > We appreciate your acknowledgment that all concerns have been fully resolved.
> > >
> > > Given this, we would like to ask whether there are any remaining issues or aspects of the work that would justify reconsidering the current score, as we believe the paper makes a novel and meaningful contribution.

---

### Official Review · Reviewer_jCzE · 2026-03-11

**Soundness:** 3
**Presentation:** 3
**Significance:** 2
**Originality:** 3
**Overall Recommendation:** 4
**Confidence:** 3

**Summary:**

This paper proposes a framework that integrates single enhancement and classification using coupled diffusion models, operating on the noisy signal and the classifier’s logits. The framework allows for mutual guidance, improving both signal recovery and classification accuracy. The authors demonstrate the effectiveness of the proposed method on image classification and automatic speech recognition tasks.

**Compliance With Llm Reviewing Policy:**

Affirmed.

**Final Justification:**

I appreciate your thoughtful response, which has satisfactorily resolved my concern on the comparision experiments. While the acceleration approach can be applied to the diffusion process, this acceleration approach can be applied to all the other methods. Therefore, the additional computation complexity still bothers me.

I would like to raise my initial score.

**Key Questions For Authors:**

1.How can the computational efficiency of the method be further optimized for large-scale real-time tasks?
2. Can the authors confirm that CARD was trained and evaluated with the same data augmentation and hyperparameter tuning as the proposed methods?

**Limitations:**

No

The authors are encouraged to discuss the computation complexity of the proposed schemes compared to baselines.

**Strengths And Weaknesses:**

Strength
1. The proposed coupled diffusion model framework combines signal denoising and logits denoising to significantly improve classification performance.
2. The method improves accuracy in both image classification and speech recognition tasks, especially under noisy conditions.


Weakness
1. While the coupled diffusion method outperforms other methods in terms of performance, it may incur higher computational costs, especially under alternating and nested strategies.
2. The comparison methods presented in the manuscript are limited.
3. The main experiments are mainly performaned on small datasets.

---

> ### Author Rebuttal · Authors · 2026-03-30
>
> Thank you for your constructive feedback. We appreciate your comments and address them below.
>
> **Answer to weaknesses:**
>
> 1.We note that this concern is directly addressed in our answer to key questions (Q1), where we discuss how the method can be made significantly more computationally efficient.
>
> 2.Additional comparison methods are addressed in our responses to other reviewers and will be incorporated into the revised version. Specifically, in response to dtF4 (Q2), we evaluated a baseline where a front-end enhancement network is trained end-to-end using the classifier loss while keeping the classifier fixed. This approach, inspired by prior works such as URIE(vision)[1] and Dissen(speech)[2], consistently underperforms our coupled diffusion framework, except for the simple case of the MNIST dataset. In addition, as discussed in dtF4 (Q3), we evaluated a non-diffusion enhancement baseline trained with a regression loss, which also yields inferior performance compared to our diffusion-based methods. Finally, following the suggestion in DZb3 (Q2), we include a scalability analysis on higher-resolution image datasets (224×224 ImageNet subset), demonstrating that the proposed framework remains stable and effective beyond low-resolution benchmarks. We will incorporate these baselines and analyses in the revised version of the paper to provide a more comprehensive and balanced comparison.
>
> 3.We would like to clarify that the image experiments included in our evaluation serve primarily as a proof of concept. They are designed to demonstrate the theoretical soundness and cross-domain applicability of the coupled diffusion framework. In contrast, our audio experiments are conducted on substantially larger and widely used benchmarks, namely EARS-WHAM and EARS-Reverb, which are part of the EARS dataset. These datasets are specifically designed for realistic speech enhancement and dereverberation, containing high-quality fullband speech with diverse noise and reverberation conditions, and are considered standard and sufficiently large-scale benchmarks in modern speech enhancement research. Moreover, EARS has already been adopted by several recent works, including “Few-step Adversarial Schrödinger Bridge for Speech Enhancement”, “Diffusion Buffer: Online Diffusion-based Speech Enhancement”, “Non-intrusive Quality Assessment with Diffusion” and more, which further supports its role as a reliable and increasingly standard evaluation benchmark. Therefore, while the image experiments are intentionally lightweight, the main evaluation of our method is performed on realistic and sufficiently large-scale speech datasets, providing a meaningful validation of our approach.
>
> **Answer to key questions:**
>
> 1.Our method can be made significantly more efficient for large-scale real-time tasks by leveraging standard diffusion acceleration techniques, without any modifications to the core framework. As discussed in our response to reviewer DZb3 (Q1), our framework is decoupled from the specific diffusion process and only defines the coupling between signal and logits. This makes it directly compatible with standard acceleration techniques for diffusion models, such as improved samplers (e.g., advanced ODE/SDE solvers), reduced-step methods, or distillation, without modifying the core method. In practice, these can significantly reduce the NFE, and we see extending the framework to real-time settings as a promising direction for future work.
>
> 2.Yes. For a fair comparison, CARD is implemented using the same logit diffusion component as in our method, with identical architecture, training data, and hyperparameters. The only difference is that CARD operates solely on the logits, without the additional signal diffusion and mutual guidance present in our approach.
>
> **Limitations:**
>
> We agree that explicitly discussing the computational complexity in the main text would improve clarity. In the current version, we do reference this analysis in the main text (Sec. 4.1 for images and Sec. 4.2 for audio), where we point to Appendix D.4 and Appendix E.5, respectively, for a detailed comparison of sampling efficiency in terms of NFE across the different strategies. However, we agree that this discussion is not sufficiently visible, and we will revise the paper to include a concise summary of the computational overhead (e.g., relative NFE across strategies) directly in Sec. 4, while keeping the full analysis in the appendix.
>
> [1] Son et al., 2020.
>
> [2] Dissen et al., 2025.

---

> > ### Author Rebuttal · Reviewer_jCzE · 2026-04-01
> >
> > I appreciate your thoughtful response, which has satisfactorily resolved my concern on the comparision experiments. While the acceleration approach can be applied to the diffusion process, this acceleration approach can be applied to all the other methods. Therefore, the additional computation complexity still bothers me.
> >
> > I would like to raise my initial score.

---

> > > ### Author Response · Authors · 2026-04-04
> > >
> > > Thank you for the constructive follow-up, and we appreciate your willingness to raise your initial score. We would like to reply to your remaining question:
> > >
> > > A key point is that logits diffusion is substantially cheaper than signal diffusion, which significantly mitigates the practical computational overhead of our method. While our framework does introduce additional cost, especially in the Alternating and Nested strategies, which are designed for accuracy-critical settings, the Parallel strategy remains relatively efficient in practice. Although it involves both signal and logits diffusion (doubling the NFE compared to standard methods), the actual runtime increase is much smaller. In our experiments on the EARS dataset, a single signal diffusion step was approximately 7× more expensive than a logits diffusion step. As a result, the added cost of logits diffusion is relatively minor, and the overall runtime does not scale proportionally with NFE. We expect this gap to persist across domains, as logits typically lie in a much lower-dimensional space than signals. We will include this clarification and empirical observation regarding computational cost in the revised manuscript.

---

### Official Review · Reviewer_dtF4 · 2026-03-12

**Soundness:** 3
**Presentation:** 3
**Significance:** 2
**Originality:** 2
**Overall Recommendation:** 3
**Confidence:** 4

**Summary:**

This paper proposes coupled diffusion models of signals and logits to improve the noise robustness of classifiers. Specifically, it integrates two interacting diffusion processes: one operating on the input signal and the other on the classifier’s output logits. This coupled formulation enables mutual guidance, where the denoised signal improves class estimation, while the evolving class logits guide the signal reconstruction toward the data manifold.

**Compliance With Llm Reviewing Policy:**

Affirmed.

**Final Justification:**

I appreciate the author's detailed response, which has partially resolved my concern about the comparison to other baselines (the remaining concern is whether the detailed training settings of the baselines are reasonable).  I would like to raise my initial score.

**Key Questions For Authors:**

See the Weaknesses section.

**Limitations:**

yes

**Strengths And Weaknesses:**

Strengths:

1)	The proposed idea is interesting and improves classifier accuracy under noisy conditions without requiring fine-tuning.

2)	The overall presentation is clear and easy to follow.

Weaknesses:

1)	The most significant concern with the proposed method is the additional latency it introduces to the classifier. As shown in Fig. 4, the denoisers require many steps to improve accuracy, except in the “Parallel” setting.

2)	Although the proposed approach does not require fine-tuning the classifier, it still needs access to all classifier weights to obtain the logits. In this case, why not fix the classifier weights and allow gradients to pass through them to train a front-end enhancement model using the classifier loss? It would be valuable to include such a comparison in the experiments.


3)	Since generative speech enhancement models may introduce hallucinations, it would also be interesting to see ASR results for a similar speech enhancement architecture trained with a regression loss instead.

4)	It is unclear why the logit enhancer also requires a generative model rather than a simpler regression-based approach.


5)	Is there any explanation why the CARD method does not work well in most cases?

---

> ### Author Rebuttal · Authors · 2026-03-30
>
> Thank you for your detailed and constructive feedback. You provided insightful comments and we would like to respond to them.
>
> **Answer to weaknesses:**
>
> 1.We agree that computational efficiency is an important consideration. Our method can be made significantly more efficient by leveraging standard diffusion acceleration techniques, without any modifications to the core framework. As discussed in our response to reviewer DZb3 (Q1), our framework is decoupled from the specific diffusion process and only defines the coupling between signal and logits. This makes it directly compatible with standard acceleration techniques for diffusion models, such as improved samplers (e.g., advanced ODE/SDE solvers), reduced-step methods, or distillation, without modifying the core method. In practice, these can significantly reduce the NFE.
>
> More broadly, using a front-end enhancement module before a downstream classifier is a well-established practice for handling  severe degradations, supported by classical and modern works[1][2][3]. Our method follows the same principle, but integrates enhancement and classification more tightly via mutual guidance. Finally, the convergence behavior in Fig. 4 further mitigates this concern: the Parallel strategy reaches strong performance in very few steps due to its step-level bidirectional feedback, while Alternating and Nested strategies delay this interaction and thus require more steps to converge (see DZb3 Q3).
>
> 2.We agree that this is an important comparison. In fact, we already conducted such experiments as part of our study. Specifically, we trained a front-end enhancement network end-to-end using the classifier loss while keeping the classifier weights fixed, running both Dissen(speech)[4] and URIE(vision)[5] on our datasets as representatives of this paradigm. This baseline consistently underperformed compared to our coupled diffusion framework, except on MNIST.
>
> Image (IN32-100, accuracy ↑, URIE vs. Parallel):
> 15% GN: 51.1 → **69.8**
> 30% GN: 26.8 → **65.5**
> Gaussian Blur: 33.6 → **63.3**
>
> Audio (WER ↓, Dissen vs. Nested):
> GCommands-Reverb: 7.96 → **4.85**
> EARS-Reverb: 6.46 → **3.83**
> EARS-WHAM: 13.04 → **9.75**
>
>  Full results are available at:
>
> https://anonymous.4open.science/r/coupled_diffusion_figures_and_tables-677A(audio_results_revised_table, image_results_revised_table).
>
> We hypothesize that while task-driven enhancement optimizes for the fixed classifier's decision boundary, it tends to introduce artifacts that satisfy the loss without robustly recovering the underlying signal. In contrast, our coupled approach maintains a stronger generative prior, leading to better generalization under noise. We will include this baseline in the revised paper for completeness.
>
> 3.We address Questions 3,4 jointly as they are highly connected. We trained a non-diffusion speech enhancer with the same architecture, optimized with a standard regression loss, on the EARS-Reverb dataset. This regression-based baseline yielded worse ASR performance compared to our diffusion-based methods. We will include it as an additional baseline in the revised version.
>
> Audio (WER ↓, EARS-Reverb, regression vs. Nested):
> 7.06 → **3.83**
>
> More broadly, the value of diffusion models for speech enhancement is well-supported in the literature, with strong results reported by SGMSE+, CDiffuSE, and recent flow and diffusion bridge approaches. These models consistently outperform regression-based counterparts, likely due to their ability to model the full distribution of clean speech rather than just the conditional mean. For the logit enhancer specifically, CARD demonstrated that applying diffusion in logit space is beneficial for conditional distribution prediction, a finding our experiments corroborate.
>
> 4.We note that this point is already briefly discussed in the Introduction of the paper. CARD often underperforms on corrupted signals because it applies diffusion within the semantic (logit) space rather than the input (signal) space. Since the logits of a corrupted signal are often distributionally distant from those of a clean signal, the denoiser lacks the necessary context to recover the original intent. Effectively, CARD is designed for fine-grained refinement of predictions, whereas corrupted inputs require structural noise removal at the signal level before any reliable semantic mapping can occur. Our method addresses this directly by jointly denoising both the signal and the logits simultaneously through mutual guidance, ensuring semantic refinement is always grounded in a progressively cleaned input signal.
>
> [1] Weninger et al., 2015.
>
> [2] Liu et al., 2022.
>
> [3] Richter et al., 2023.
>
> [4] Dissen et al., 2025.
>
> [5] Son et al., 2020.

---

> > ### Author Rebuttal · Reviewer_dtF4 · 2026-04-02
> >
> > I thank the authors for their response and clarifications. It solves part of my concerns. However, I still have the following questions:
> >
> > 1) In my previous comment: "In this case, why not fix the classifier weights and allow gradients to pass through them to train a front-end enhancement model using the classifier loss?...".  In the literature, researchers typically first train a baseline enhancement model (e.g., using a regression loss) and then **fine-tune** it jointly with a classifier loss (while keeping the classifier weights fixed), rather than directly inserting a randomly initialized module before the classifier. It would be very interesting to see the results for both the baseline enhancement model and its fine-tuned version.
> >
> > 2) Diffusion models for speech enhancement are good at improving quality but not for preserving fidelity (e.g., content hallucinations). Previous works [1] show that its WER is usually worse than that of the regression model.
> >
> > [1] Chhaglani, B., Gao, Y., Richter, J., Li, X., Zadissa, S., Pruthi, T., & Lovitt, A. (2025). "ArtiFree: Detecting and Reducing Generative Artifacts in Diffusion-based Speech Enhancement."

---

> > > ### Author Response · Authors · 2026-04-04
> > >
> > > We thank the reviewer for their response and would like to further clarify the remaining concerns.
> > >
> > > 1.For the baseline enhancement model, we trained a regression-based enhancer on the EARS-Reverb dataset, which yielded inferior performance compared to our coupled diffusion methods (e.g., WER: 7.06 → 3.83). As noted in our previous response, we will extend this baseline to all datasets and include the full comparison in the revised manuscript. Regarding fine-tuning with classification loss, We would like to clarify an important detail about the training procedure of the baselines we compared against. URIE (vision) is not trained using classification loss alone, it is trained jointly on both enhancement (regression) and classification losses simultaneously. Similarly, Dissen (audio) first undergoes a regression-loss warmup phase, and is subsequently trained on a combination of regression and classification losses, with the weight of the classification loss gradually increasing over the course of training. The results reported in our previous response reflect this combined training regime, not a classification-loss-only setup. Despite following this protocol, these methods still underperform compared to our coupled diffusion framework across datasets. We will clarify this training procedure more explicitly in the revised version to avoid any confusion.
> > >
> > > 2.We agree that content hallucinations are a well-documented limitation of standard diffusion-based speech enhancement, as highlighted by ArtiFree and related work. However, we respectfully argue that this concern does not directly apply to our framework, for the following reasons:
> > >
> > > a. Our approach is fundamentally different from standard diffusion-based Speech Enhancement.
> > > The artifacts discussed in ArtiFree arise in standalone diffusion enhancement models (such as SGMSE+) that operate independently of any downstream task. In our coupled diffusion framework, the diffusion process is not used as a standalone front-end, it is tightly coupled with the classifier. The generative process is guided by the classifier's own representation, which acts as a semantic prior that steers the output toward content-preserving reconstructions. This is precisely the type of semantic grounding that ArtiFree itself identifies as the key to reducing artifacts.
> > >
> > > b. Our empirical results directly contradict the concern in our setting.
> > > While prior works have reported that diffusion-based speech enhancement may degrade ASR performance due to fidelity issues (e.g., hallucinations), our empirical results do not reflect this behavior. In our experiments, our coupled methods consistently outperform both regression-based enhancement and regression models fine-tuned with classification loss. Specifically, a regression-only enhancer trained on EARS-Reverb yields worse WER compared to our method (7.06 → 3.83). Moreover, when following the standard paradigm of combining regression and classification losses (e.g., Dissen), performance still remains inferior to our approach across datasets (e.g., EARS-Reverb: 6.46 → 3.83, EARS-WHAM: 13.04 → 9.75). This indicates that, in our setting, diffusion improves rather than degrades semantic fidelity. We attribute this to our formulation, where denoising is guided by semantic information through the classifier, reducing hallucinations.
> > >
> > > In summary, while the concern is valid for unconstrained diffusion enhancement, our coupled design addresses it structurally, and our WER results confirm this empirically.

---

### Official Review · Reviewer_4Mhg · 2026-03-13

**Soundness:** 3
**Presentation:** 2
**Significance:** 3
**Originality:** 3
**Overall Recommendation:** 4
**Confidence:** 3

**Summary:**

The paper

- tackles the problem of classification when signals are degraded
- Proposes to couple enhancement and recognition and jointly train of two interacting diffusion models: one over the signal and one over classifier logits
- validate the method on image classification and automatic speech recognition

**Compliance With Llm Reviewing Policy:**

Affirmed.

**Final Justification:**

- I thank the authors for their rebuttal
- My main concerns have been addressed
- Overall, I find the work solid, and with the revisions the authors have committed to making, the paper leaves a positive impression
- That said, after reading the rebuttal, in particular the discussion with Reviewer DZb3, I do not feel able to raise my score beyond my initial recommendation of weak accept

**Key Questions For Authors:**

- Ablations: the parallel strategy reaches high accuracy in very few steps ($\sim 5$) and then saturates, unlike the other strategies. Why does parallel inference behave this way? similarly, can the authors comment on why does the alternating strategy scale poorly and underperform?
- Can you summarize practical pros/cons of the three strategies (when to choose each in terms of compute, stability, and accuracy)?
- Since classification is discrete, have you considered replacing continuous logit diffusion with a discrete diffusion process over labels (to potentially leverage ground-truth class labels?)

**Limitations:**

Performing diffusion on logit maybe fragile as choice: it depends on access to classifier logits and ties the approach to a particular pretrained classifier/semantic space and doing so prevent the model from using ground-truth labels.
In the same vein, the framework appears not to leverage ground-truth class labels directly (even when available at training time), this may potentially induce a bias: the bias of the pre-trained classifier would leak to the trained model

**Strengths And Weaknesses:**

The model specification and conditional structure is unclear
- The graphical model/dependencies among $x_{0:T}$, $y_{0:T}$, $x_{\text{corr}}$, and $y_{\text{corr}}$ are not clearly defined.
- This makes some simplifications feel inconsistent; e.g., Col. 1, lines 234--236 drop dependence on $x_t$ in favor of $x_{t-1}$, but an analogous simplification is not applied elsewhere (e.g., Eq. 10 keeps both $y_0$ and $y_t$ in the transition).

Confusion in some of the equations:
- Equation 9 and 10: in standard diffusion models x_0^{(t)} would have been the expectation of the the denoising posterior distribution $p(x0 |x_t,y_t,xcor)$,
while in equation 10 $x_0^{(t)}$ is rather sampled from  p(x0 |xt,y^{(t+1)}_0 ,y_t,xcor) which condition on the estimate $y^{(t+1)}_0$ as well? same for equation 13 and 14.

Minor Issues
- In Algorithms 1 and 3: avoid using $\sim$ for deterministic estimates such as $\hat{x}_0^{(t)}$ and $\hat{y}_0^{(t)}$ (e.g., Algo. 1, line 6; Algo. 3, line 9).
- "ASR" is used before being defined see Col 1. Lines 78
- Define $\tilde{\beta}$ in Eq. (2) and cite a standard choice (there are multiple choices).
- Col. 1, lines 122--128: several statements are misleading: (i) the denoiser does not define the exact $p_{0\mid t}(x_0 \mid x_t)$, which is generally intractable.
(ii) There is not a one-to-one mapping between $x_0$ and $x_t$: the forward noising induces a lost of information, so a single $x_t$ can correspond to many possible $x_0$.

---

> ### Author Rebuttal · Authors · 2026-03-30
>
> We thank the reviewer for the detailed and constructive feedback. We address each point below.
>
> **1. Model specification and dependencies**
>
> Figure 2 is intended as a high-level schematic. The precise dependencies are specified in detail within the algorithmic descriptions. If there is a particular dependency or case that remains unclear, we would appreciate a more specific pointer so we can clarify it directly. Regarding the inconsistency, In Eq. 10, $\hat{y}_0^{(t+1)}$ is not the true clean target but the model's estimate. Since this estimate is imperfect, we retain the dependence on $y_t$, whereas if it were the true $y_0$, the dependence could be dropped analogously. We will clarify this in the revision.
>
> **2. Clarification of Equations (9–10, 13–14)**
>
> We agree the notation can be improved. The estimate is deterministic (not sampled), and we will remove the use of $\sim$. Regarding the conditioning on $\hat{y}_0^{(t+1)}$, since this estimate is available from the previous step, we leverage it as additional conditioning to improve the denoising prediction. The same applies to Eqs. 13–14, which we will clarify in the revision.
>
> **3. Minor issues and clarifications**
>
> We will:
> - Define ASR at first occurrence and specify $\beta_t$ (standard noise schedule).
> - Clarify that the denoiser estimates $\mathbb{E}[p(x_0|x_t)]$, not the full posterior. Regarding the mapping, our discussion concerns $x_{t-1}$ rather than $x_t$. The key point is that different parameterizations ($x_{t-1}$, $\hat{x}_0^{(t)}$, or $\hat{\epsilon}_t$ the noise) are equivalent given $x_t$, as one can be recovered from another. For example $\hat{\epsilon}_t = \frac{x_t - \sqrt{\bar{\alpha}_t} \hat{x}_0^{(t)}}{\sqrt{1 - \bar{\alpha}_t}}$ where alpha is part of the noise schedule.
>
> **4. Key questions**
>
> 1.The Parallel strategy's fast convergence is addressed in our response to Reviewer DZb3, Q3. The Alternating strategy underperforms because it conditions each diffusion block on a single fixed endpoint estimate, committing the model to one guess with no uncertainty signal. Parallel and Nested instead use intermediate estimates, which carry inherent uncertainty that allows adaptive weighting, less early on, more as estimates stabilize. This acts as a regularizer for faster error correction. Fixed endpoint conditioning can work in simple settings but becomes a bottleneck in complex ones, explaining the slower convergence and weaker performance seen in our results.
>
> 2.Each strategy reflects a different trade-off between compute, stability, and accuracy. All costs (×2, ×7, ×10) are relative to signal-only diffusion.
>
> (1) Parallel exchanges intermediate estimates at every timestep, enabling fine-grained coupling, fast convergence, and strong accuracy, especially on complex datasets, at a relatively low computational cost (≈2×).
>
> (2) Alternating uses fixed endpoint conditioning per diffusion block, making it simple and stable, but coarse and less adaptive on complex data, as discussed in Q1 of key questions, and is the most computationally expensive (≈10×).
>
> (3) Nested recomputes the signal for each logit timestep, yielding stronger robustness and accuracy in challenging settings, but at significant cost (≈7×). In practice, Parallel is a strong default, Alternating suits simpler and more stable regimes, and Nested is preferable when accuracy is prioritized over compute. We will incorporate this discussion in the revised paper.
>
> 3.Our choice of continuous logit diffusion is motivated by the need for a rich semantic representation that can effectively guide signal denoising. Logits provide a continuous, structured space capturing class similarities and uncertainty, whereas discrete labels are low-dimensional and convey no information about inter-class relationships or confidence. Furthermore, our bidirectional coupling relies on representations that evolve smoothly and can be conditioned upon at intermediate steps, which is natural in continuous logit space but less straightforward in a discrete setting. That said, discrete diffusion over labels is a promising direction, potentially allowing stronger use of supervision or label priors, and we consider it an interesting avenue for future work.
>
> **Limitations:**
>
> Our goal is to enhance the input signal to better align with the classifier's existing representation under corruption, not to modify its decision boundaries. While our method inherits the pretrained classifier's biases, this is true of any approach using a fixed classifier. We keep it frozen to avoid overfitting to limited, corruption-biased data and preserve generalization. Our framework can also be trained jointly with the classifier, but we intentionally avoid this to maintain general applicability, including unlabeled settings. Incorporating ground-truth labels or joint training is possible and represents an interesting direction for future work, but would shift the focus toward classifier adaptation, which is beyond our scope.

---

> > ### Author Rebuttal · Reviewer_4Mhg · 2026-04-01
> >
> > - I thank the authors for their response and clarifications.
> >
> > - To clarify my earlier comment under "Minor issues," specifically point 3, bullet 2 in your answer: the variance of the approximate reverse transition can be defined in at least two standard ways. It can be taken as the variance of the forward transition from $t$ to $t+1$; see eq. (2) and the fifth line preceding it in Wu et al. (2024) [1]. Alternatively, it can be taken as the variance of the transition at time $t$ conditioned on times $0$ and $t+1$; see eq. (7) in Ho et al. (2020) [2]. Just mention the one you are using for clarity. My understanding is that the current formulation follows DDPM.
> >
> > - While Figure 2 is useful for illustrating dependencies, it is better described as an architectural schematic with three configurations rather than as a graphical model. A graphical model should formally justify the conditional dependencies between variables and explain why some conditioning variables are omitted in favor of others; see, Chapter 8 of [3].
> >
> > - In light of the authors response, I also believe the limitations should more explicitly acknowledge that the approach may inherit biases from the classifiers it relies on.
> > - Likewise, given the way the method is constructed, it does not appear able to exploit ground-truth labels directly, and this should also be stated as a limitation.
> >
> > ---
> >
> > .. [1] Wu, Luhuan, et al. "Practical and asymptotically exact conditional sampling in diffusion models." Advances in Neural Information Processing Systems 36 (2024) https://arxiv.org/pdf/2306.17775
> >
> > .. [2] Ho, Jonathan, Ajay Jain, and Pieter Abbeel. "Denoising diffusion probabilistic models." Advances in neural information processing systems 33 (2020): 6840-6851.
> >
> > .. [3] Bishop, Christopher M. Pattern recognition and machine learning. New York: springer, 2006.
> > https://www.microsoft.com/en-us/research/wp-content/uploads/2006/01/Bishop-Pattern-Recognition-and-Machine-Learning-2006.pdf

---

> > > ### Author Response · Authors · 2026-04-04
> > >
> > > We thank the reviewer for this helpful clarification and will revise the manuscript accordingly; specific changes are outlined below.
> > >
> > > 1.The reviewer’s second option is indeed correct in our case: we follow the DDPM parameterization of Denoising Diffusion Probabilistic Models, where the reverse variance is given by the posterior variance​.
> > >
> > > $\tilde\beta_t = ((1-\bar\alpha_{t-1})/(1-\bar\alpha_t))\beta_t$
> > >
> > >  We will make this explicit in the revised manuscript.
> > >
> > > 2.We agree that Figure 2 should be described as an architectural schematic illustrating the three configurations, rather than a formal graphical model. In the revised manuscript, we will update the terminology and clarify the figure's role.
> > >
> > > 3.We agree that both aspects, the potential inheritance of biases from the pretrained classifier and the lack of direct use of ground-truth labels, should be more explicitly stated as limitations, and we will revise the manuscript accordingly.

---

### Decision · Program_Chairs · 2026-04-30

**Decision:**

Accept (regular)

**Comment:**

The paper propose an interesting acceleration approach for diffusion models. Some concerns were raised by the reviewers and most of them were addressed adequately by the authors.